# Probabilistic day-ahead forecasting of system-level renewable energy and electricity demand

Guillermo Terrén-Serrano [1,2] ✉, Ranjit Deshmukh [1,2,3] & Manel Martínez-Ramón[4]

Increasing shares of wind and solar generation, together with rising electricity demand, introduce growing uncertainty into power system operations. Accurate day-ahead forecasts of electricity demand and renewable generation are essential for system operators to coordinate electricity markets and maintain reliability at low cost. Here, we show that forecasting based on joint probability distributions of demand and renewable supply can substantially improve system-level forecasting performance using publicly available weather data. We develop multiple day-ahead forecasting models that combine machine learning methods to identify relevant weather variables with probabilistic approaches to quantify forecast uncertainty, and we evaluate these models using proper scoring rules. Applied to the three zones of the California Independent System Operator, the best-performing model improves forecast skill by 25% relative to current benchmarks. We further show that forecasts based on joint probability distributions enable a more effective allocation of operating reserves than conventional deterministic approaches, highlighting the potential of probabilistic machine learning to enhance market efficiency and grid stability in increasingly decarbonized power systems.

Clean energy targets and falling technology costs are driving a rapid increase in the share of wind and solar generation, also known as Variable Renewable Energy (VRE), in electricity systems[1,2]. At the same time, electricity demand is growing due to air conditioning adoption to mitigate the increase in average temperatures[3], growing use of electric vehicles[4], and increasing deployment of data centers[5]. Accurately forecasting demand and supply at hourly and sub-hourly resolution is essential for power system operators to commit adequate generation, storage, and demand resources a day ahead of the actual dispatch in order to maintain reliability of the electricity system. However, increasing electricity demand and weather-dependent energy sources, as wind and solar, add variability and uncertainty to both demand (Fig. 1a–c) and supply (Fig. 1d–h), increasing the challenges for power system operators to forecast these resources. These challenges are

even more intensified under the effects of climate change and extreme weather conditions[6].

In the United States, Independent System Operators (ISOs) or Regional Transmission Organizations (RTOs) operate wholesale electricity markets, manage the power grid, and serve about 70% of electricity demand, the rest of which is served by vertically integrated utilities[7]. The main goal of the power system operators is to balance electricity demand and supply to ensure the reliability of the electricity system. Through the day-ahead wholesale electricity market, the system operators schedule the least-cost generators, subject to power flow constraints, based on day-ahead forecasts of electricity demand and generator (and storage) availability[8]. To account for day-ahead forecast errors, the system operators run a real-time market an hour or two before real-time dispatch. In the case of CAISO, the real-time market has

[1]Environmental Studies, University of California Santa Barbara, Santa Barbara, CA, USA. [2]Environmental Markets Lab (emLab), University of California Santa Barbara, Santa Barbara, CA, USA. [3]Bren School of Environmental Science and Management, University of California Santa Barbara, Santa Barbara, CA, USA. [4]Department of Electrical and Computer Engineering, University of New Mexico, Albuquerque, NM, USA. ✉e-mail: guillermoterren@ucsb.edu

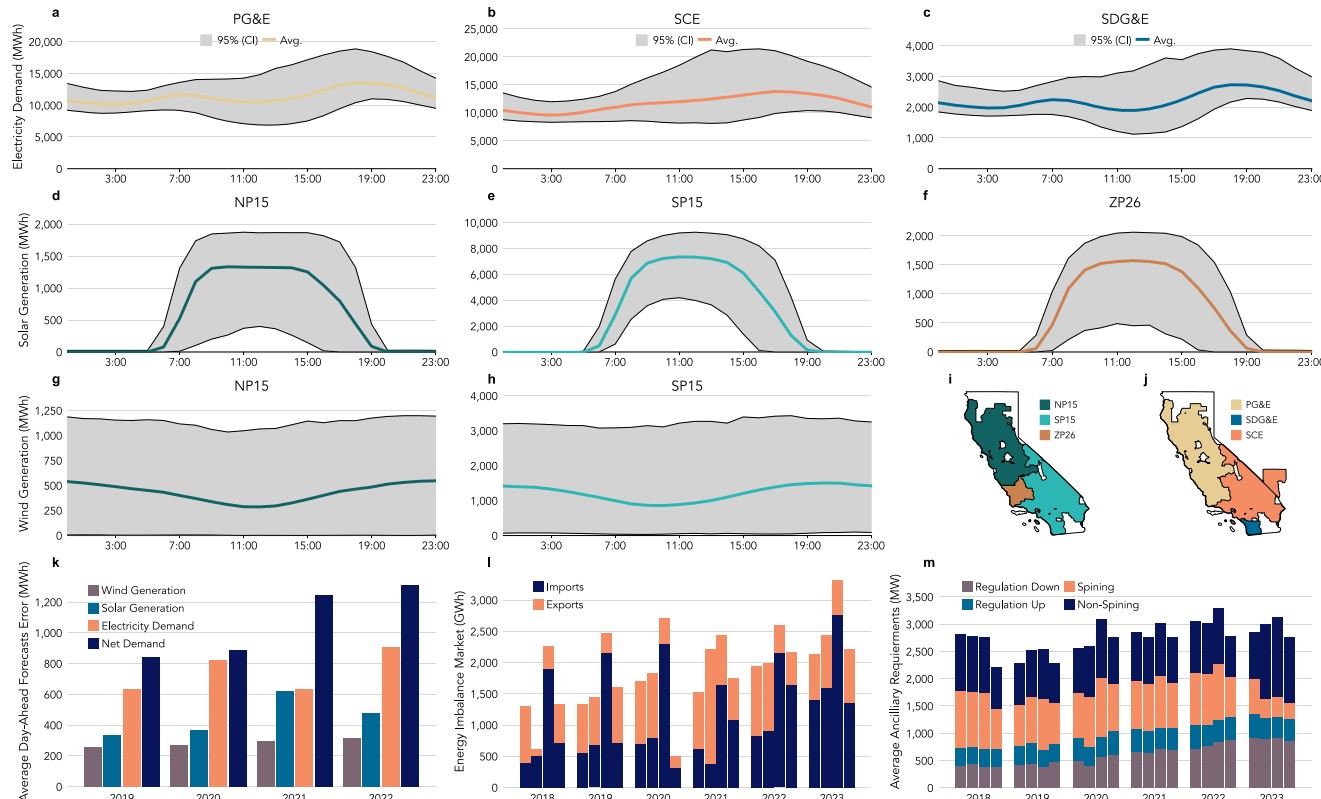

**Fig. 1 | Raising day-ahead forecast errors from electricity demand and renewable generation increases operating requirements and area control errors.** Electricity demand variability experienced by California's three main Investor-owned Utilities, Pacific Gas and Electric (PG& E) (**a**), Southern California Edison (SCE) (**b**), and San Diego Gas and Electric (SDG& E) (**c**). Solar generation variability in trading zone NP15 (**d**), SP15 (**e**), and ZP26 (**f**). Wind generation variability in the trading zone NP15 (**g**) and SP15 (**h**). The gray regions represent the spread of the 95% Confidence Interval (CI), and the lines are the average hourly demand or generation in 2020. **i** The map show the approximate boundaries of the zones managed by CAISO (adapted from oasis.caiso.com). **j** Areas served by three major utilities (PG& E, SCE, and SDG& E) in CAISO's electricity market. **k** Hourly errors in the CAISO's day-ahead forecast for wind and solar generation and electricity demand, and net demand. **l** Imports and exports from CAISO in the energy imbalance market. (**m**) Average annual ancillary services capacity requirements for regulation up and down, spinning, and non-spinning reserves in CAISO. Regulation down and non-spinning reserves had a major increase, while spinning reserves were the only product that decreased in 2023.

evolved into an Energy Imbalance Market (EIM) that spans across several regions in the Western United States, which was designed to reduce costs by allowing access to many more generation resources beyond California. To address the remaining differences between committed resources in the real-time market and actual demand, as well as for contingency events such as generator or transmission outages, the power system operator also procures operating reserves in the day-ahead market. Operating reserves are additional capacity purchased from committed generators to compensate for forecasting errors[9] and ensure reliability during contingency events[10]. As the share of weather-dependent generation has grown, aggregated system-level forecast errors have also grown (see CAISO day-ahead forecast errors of wind and solar generation, electricity demand, and net demand or the difference between demand and wind and solar generation in Fig. 1k). Larger forecast errors lead to significant changes in the generation, storage, and transmission scheduling close to real-time dispatch[11], which has increased the electricity exchange in the energy imbalance market[12] (see imports and exports in CAISO's Energy Imbalance Market in Fig. 1l), and the operating reserve requirements necessary to ensure a reliable electricity supply[13,14] (see CAISO's reserves or Ancillary Services requirements in Fig. 1m), both of which have increased operating costs[15].

The additional costs associated with operating reserves (Supplementary Fig. 1), procured in the day-ahead ancillary services markets, are ultimately borne by electricity ratepayers[16]. The generators that respond the fastest to provide upward operating reserves are often

natural gas combustion turbines, which are expensive and have higher greenhouse gas and criteria emissions compared to other technologies[17]. Storage, including battery technologies that also provide operating reserves, may also have greenhouse gas and criteria emissions associated with charging energy and energy losses, depending on the grid conditions during charging. Failure to compensate for forecast errors through scheduling changes and operating reserves leads to area control errors, variations in system frequency, and blackouts in severe cases[6]. Therefore, reducing the errors of the electricity supply and demand forecast is critical to minimize costs[18], emissions, and reliability issues[19].

Addressing these challenges introduced by the uncertainty in the electricity demand and the VRE generation requires characterizing the relation between weather and energy accurately[20], identifying the information sources from Numerical Weather Forecasts (NWFs) that improve day-ahead energy forecasts[21]. Furthermore, probabilistic forecasts could enable us to determine operating reserve levels dynamically based on the uncertainty in the prediction, which may reduce the requirements for and costs of operating reserves.

Previous research focused on Machine Learning (ML) to improve electricity demand and VRE generation forecasts from NWFs instead of using physical models[22]. NWFs have numerous weather features, so identifying the most informative variables is essential to reduce collinearity[23]. The discovery of patterns leads to an increase in the effectiveness of a forecast[24]. Deep learning methods learn patterns from high-dimensional data with spatial structures and efficiently

deal with collinearity, but they require substantial amounts of data[25–28]. With fewer observations, structured sparsity regularization methods efficiently uncover spatial patterns and reduce the dimensionality of input features. More recently, deep learning methods based on Temporal Fusion Transformers[29,30], Informers[31], and TimesNets[32] were used in energy forecasts. However, the proposed methodologies generally forecast a single energy feature (demand, solar, or wind) and do not provide a predictive multivariate density function to draw predictive scenarios that preserve the time structure in risk assessment applications.

Probabilistic day-ahead energy forecasts at the asset level based on pattern similarity improved on Bayesian forecasts[33] by adding time correlation between intervals to generate predictive scenarios[34,35]. Both studies assume a uniform relation between the input weather features to forecast a single resource, but they do not consider collinearity reduction and the joint nature of weather-dependent resources. System-level forecasts are less researched despite their role in determining operating reserve requirements. In addition, asset-level demand and forecasts[28,36,37] generally utilized the open-source NWFs provided by the European Centre for Medium-Range Weather Forecast (ECMWF), which has a 9 × 9 km spatial resolution[38], or historical data. Yet, the High-Resolution Rapid Refresh (HRRR) NWFs is also publicly available and provide continental-scale NWFs with a temporal resolution of 1 h and 3 × 3 km spatial resolution[39,40], superior to global-scale NWFs[41,42].

In this study, we show that joint probabilistic day-ahead forecasts of electricity demand and wind and solar generation improve system-level uncertainty characterization using data from the HRRR NWF. We develop a probabilistic ML framework[43] that combines sparse feature selection[44] with multi-task Gaussian Process (GP) regression[45,46] to jointly model electricity demand, solar generation, and wind generation at the system level. The approach produces full predictive density functions and time-consistent predictive scenarios, enabling a probabilistic assessment of forecast uncertainty and operating reserve requirements. We apply the proposed methodology encompassing processing, modeling, and model selection (Supplementary Fig. 2), to the electricity system operated by California Independent System Operator (CAISO), producing hourly day-ahead forecasts for three nodal regions (Fig. 1i)−Northern (NP15), Southern (SP15), and Central (ZP26) California−and major load-serving utilities (Fig. 1j)−Pacific Gas & Electric (PG&E), Southern California Edison (SCE), and San Diego Gas & Electric (SDG&E). Using multivariate proper scoring rules[47], we evaluate multiple combinations of sparse and Bayesian learning methods (4 sparse × 4 Bayesian, plus 4 joint models at the regional and nodal level) and demonstrate improved forecast calibration and skill relative to existing approaches. These results highlight the value of joint probabilistic forecasting for improving operational planning, reserve allocation, and reliability in electricity systems with high shares of variable renewable energy.

## Results

### AI-based probabilistic models enhance the performance of a day-ahead energy forecast

Deterministic and probabilistic forecasts differ fundamentally. Probabilistic forecasts, specifically Bayesian forecasts, predict a density function, whereas deterministic forecasts predict point estimates. However, the predictive mean of probabilistic forecasts could be compared with the point estimates of deterministic forecasts. In this study, we compare the results of the proposed probabilistic forecasts with three different deterministic forecasts−the persistence (naive), the climatology (autoregressive), and reference forecasts (CAISO), which is a standard practice in the day-ahead forecasting literature[48].

In the first step of our analysis, we identify the reference forecast, which is the baseline forecast with the lowest Root Mean Squared Error (RMSE), to compare with the proposed forecast (see the forecast's

operational characteristics in Fig. 2i). We define the day-ahead forecast for each energy feature $\widehat{y}$−electricity demand ($\mathcal{L}$), solar ($\mathcal{S}$) and wind ($\mathcal{W}$)−at a node $z$ as $\widehat{y}_{t,z}$, where $t$ is the hour of the day (1, …, 24) and $z$ corresponds to each node (NP15, SP15, ZP26). The system-wide forecast can then be estimated by aggregating the nodal forecasts as $\widehat{y}_t = \sum_{z=1}^{Z} \widehat{y}_{t,z}$. Similarly, we estimate day-ahead forecasts for net demand ($\mathcal{N}$) as $\widehat{y}_{t,z}^{\mathcal{N}} = \widehat{y}_{t,z}^{\mathcal{L}} - \widehat{y}_{t,z}^{\mathcal{S}} - \widehat{y}_{t,z}^{\mathcal{W}}$ at the nodal level (NP15, SP15, and ZP26), and $\widehat{y}_t^{\mathcal{N}} = \sum_{z=1}^{Z} \widehat{y}_{t,z}^{\mathcal{L}} - \widehat{y}_{t,z}^{\mathcal{S}} - \widehat{y}_{t,z}^{\mathcal{W}}$ at the system level. We then evaluate RMSE over all hours (Supplementary Note 1) and normalize by the mean target value $\bar{y} = \frac{1}{KT} \sum_{k,t} y_{k,t}$, $k$ is the day and $K$ is the number of days in the testing set. CAISO forecast has the lowest Normalized RMSE (NRMSE) for electricity demand (4.3%), solar (23.2%), and wind generation (16.3%); see Fig. 2a. Similarly, CAISO day-ahead forecasts for net demand have the lowest NRMSE at NP15 (6.6%), SP15 (13.1%), and ZP26 (15.1%); and system level (8.6%); see Fig. 2b. Note that ZP16 does not have wind resources.

In the second step, we compare our proposed day-ahead forecast to the CAISO forecast (reference). The Skill Score (SS) assesses improvements in the RMSE (Supplementary Note 1), resulting in a different $SS_{RMSE}$ for each combination of sparse and Bayesian methods (Fig. 2a, b). Our proposed sparse methods, Lasso, Orthogonal Matching Pursuit (OMP), Elastic Net (EN), and Group Lasso (GL) have different formulations (see Section Sparse Learning). The objective is to identify the most effective regularization, which forces the model to discover simpler patterns in the input feature vectors. The feature vectors for electricity demand ($\mathbf{x}_i^{\mathcal{L}}$), solar ($\mathbf{x}_i^{\mathcal{S}}$) and wind generation ($\mathbf{x}_i^{\mathcal{W}}$) are from the reanalysis dataset ($\mathcal{A}$); see in the Feature Vectors for Sparse Learning.

The Bayesian learning methods explore different data assumptions: Bayesian Linear Regression (BLR) assumes linearity, Relevance Vector Machine (RVM) emphasizes sparsity, Gaussian Process Regression (GPR) accounts for non-linearity, and Multi-Task GPR (MTGPR) models joint distributions among response variables. In particular, the System-Level MTGPR (SLGPR) assumes a joint distribution across nodes (NP15, SP16, or ZP26) for the independent energy features (electricity demand, solar or wind generation); see Fig. 2c, d. In addition, the Node-Level MTGPR (NLGPR) assumes a joint distribution across energy features for the independent nodes (Fig. 2e, f). The non-linear properties come from mapping the feature vectors to high-dimensional space with a kernel function (Supplementary Note 2). The formulations include the model chain to preserve the time structure (see Sections Bayesian learning and Model Chain). The graphical representations of the algorithms are in Supplementary Fig. 3. The feature vectors are from the forecasts dataset ($\mathcal{F}$) for electricity demand ($\widehat{\mathbf{x}}_{k,t}^{\mathcal{L}}$), solar ($\widehat{\mathbf{x}}_{k,t}^{\mathcal{S}}$), wind generation ($\widehat{\mathbf{x}}_{k,t}^{\mathcal{W}}$), and joint demand and generation ($\widehat{\mathbf{x}}_{k,t}^{\mathcal{E}}$); see in the Pattern Vectors for Bayesian Learning. These feature vectors differ from those used in the sparse learning step. The hyperparameters, which control the different aspects of the learning process in the sparse and Bayesian learning methods, are cross-validated jointly (see Section Experimental Setup).

The proposed forecasts improve over the CAISO forecasts for each energy feature and net demand at the nodal and system levels (see results in Supplementary Tables 1 and 2). The models with higher $SS_{RMSE}$ with independent energy features at the system level are Lasso-SLGPR (6.1%) and EN-SLGPR (16.8%) with a linear kernel ($\mathcal{K}_L$) for electricity demand and solar generation, and EN-SLGPR (5.9%) with Matérn kernel and parameter $v = 2.5$ ($\mathcal{K}_{M_{2.5}}$) for wind generation (Fig. 2c, e). The models with higher $SS_{RMSE}$ with independent nodes are EN-BLR (19.4%) at NP15, OMP-BLR (16.7%) at SP15, and OMP-NLGPR with $\mathcal{K}_L$ (10.1%) at ZP26 (Supplementary Fig. 2d,f). EN-BLR (25.2%) and Lasso-NLGPR with $\mathcal{K}_L$ (24.8%) have the highest $SS_{RMSE}$ at the system level (CAISO) when assessing a model across nodes or energy features, respectively (Fig. 2f).

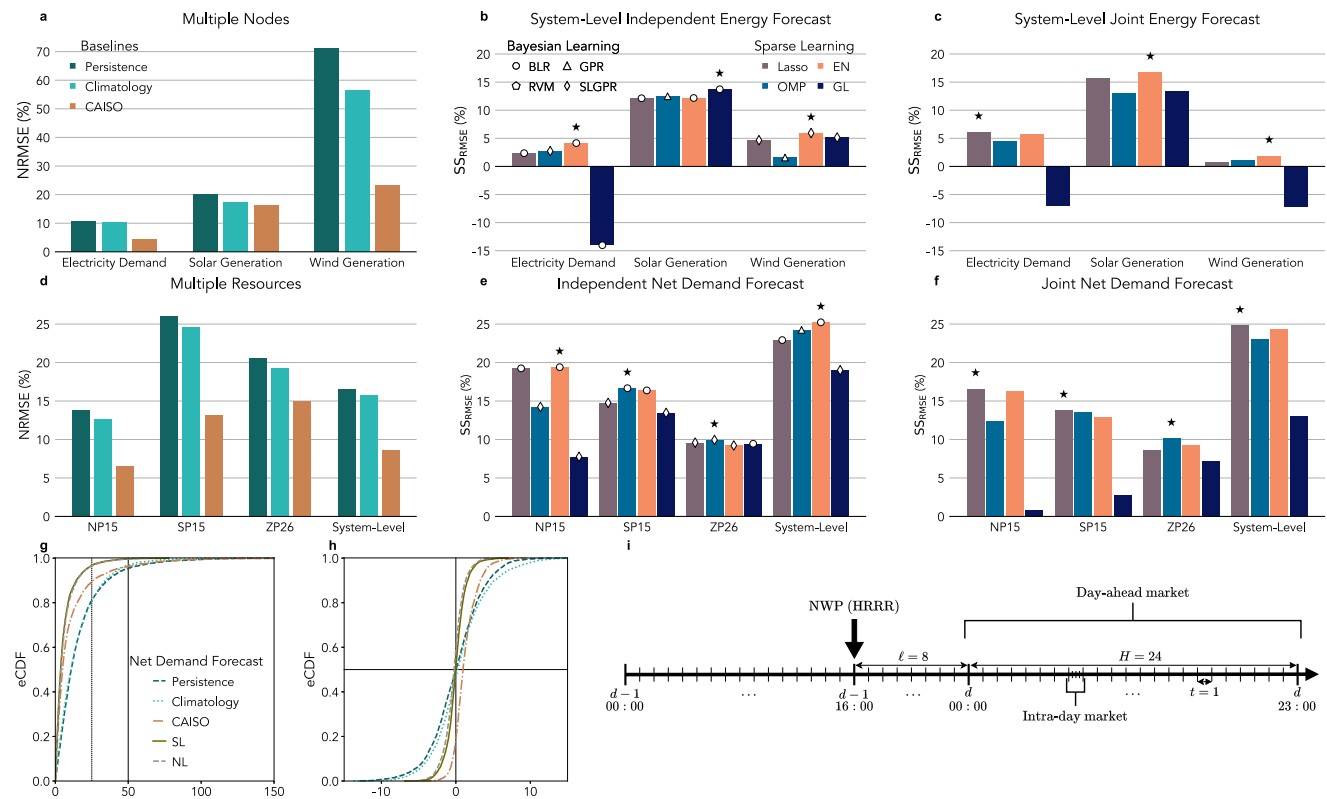

**Fig. 2 | Performance comparisons between day-ahead probabilistic and deterministic baseline forecasting models.** Normalized Root Mean Squared Error (RMSE) achieved by the baseline models (NRMSE lower is better). The baselines are the persistence, climatology, and CAISO forecasts. **a** System-level day-ahead forecast NRMSE for each energy feature (electricity demand, solar, and wind generation), and **b** net demand forecast NRMSE at each node (NP15, SP16, ZP26) and at the system level. Forecast Skill Scores (SS) for (**b**) independent and (**e**) joint day-ahead forecasts for each energy feature aggregated across nodes (SS higher is better), and net demand (**e**) independent and (**f**) joint probabilistic forecast SS at nodal and system levels (⋆ marks the model with highest SS). The colors vary by sparse

learning methods: Lasso, Orthogonal Matching Pursuit (OMP), Elastic Net (EN), and Group Lasso (GL). The markers denote the Bayesian methods: Bayesian Linear Regression (BLR), RVM (Relevance Vector Machine), GPR (Gaussian Process for Regression), and System-Level GPR (SLGPR). Empirical Cumulative Distribution Function (eEDF) of the absolute percentage residual value (**g**) and the residuals (**f**) for the day-ahead system-level net demand forecast from the persistence, the climatology, CAISO, and the best node-level (NL) ensemble and the best system-level (SL) ensemble. **i** Proposed day-ahead forecast for day $d$ submitted in $d − 1$ with lag ($\ell = 8$ h) and horizon ($H = 24$ hours). NWF (HRRR) is the Numerical Weather Forecast High-Resolution Rapid Refresh model.

In the third step, we analyze the day-ahead net demand forecast residuals at the system level. The empirical Cumulative Distribution Function (eCDF) of absolute percentage residuals gives a sense of the probability of large forecasting errors (5833 test samples); see Fig. 2g. Day-ahead forecasts using persistence or climatology, which do not include information from NWF, have large errors. CAISO's forecast significantly reduces the probability of errors greater than 25% in the net demand (0.1), while our SLGPR and NLGPR models reduce it even further (0.05). Furthermore, errors over 50% of net demand from the SLGPR and NLGPR are almost negligible when the CAISO forecast has a similar probability to the persistence and climatology (0.05). The residual eEDF indicates a potential bias in the CAISO forecast to overestimate net demand (Fig. 2h). When calculating the absolute residual statistics for each day hour (243 test samples), SLGPR and NLGPR have a low median at all hours, and the CAISO forecast has the lowest median from 6 pm to 9 pm (Supplementary Fig. 4). The absolute residual statistics calculated for the different net demand percentiles (5832 samples) and CAISO have the highest median in the 1st percentile and the lowest in the 10th percentile (Supplementary Fig. 5).

### Model selection of a probabilistic day-ahead energy forecasts with independent energy features

A joint forecast is often more desirable because it captures dependencies between response variables. Multivariate proper scoring rules assess the advantage of a joint distribution between nodes

(Supplementary Note 1). The proposed scoring rules are the Energy Score (ES), Variogram Score with $p = 0.5$ (VS$^{0.5}$), and Interval Score (IS). Each score evaluates a different property of a probabilistic forecast, so assessing multiple scores is necessary to select a balanced model[47].

The ES assesses how well the predictive scenarios $\widehat{y}_t^\star = [y_{t,1}^\star \cdots y_{t,z}^\star]$ from the predictive distribution $\widehat{y}_t^\star \sim \mathcal{N}(\widehat{\mu}_t^\star, \widehat{\Sigma}_t^\star + \mathrm{diag}(\sigma_{n_t}^2))$ represent a predictive distribution. VS$^{0.5}$ evaluates how well the scenarios preserve the time structure. IS evaluates the intervals derived from the predictive density $\mathcal{N}(\widehat{\mu}_{t,z}^\star, \widehat{\sigma}_{t,z}^{2\star})$ at different confidence levels (60%, 80%, 90%, 95%, and 97.5%). IS is not a multivariate score, so it is evaluated aggregated across nodes $\widehat{y}_t^\star = \sum_z \widehat{y}_{t,z}^\star$, and only with variances $\widehat{\sigma}_{t,z}^{2\star} = \mathrm{diag}(\widehat{\Sigma}_t^\star) + \sigma_{n_{t,z}}^2$ obtained from the predictive covariance $\widehat{\Sigma}_t^\star + \mathrm{diag}(\sigma_{n_t}^2)$.

The most suitable model for a probabilistic day-ahead electricity demand forecast is EN-BLR (see results in Supplementary Table 3). This model has the lowest ES (39.3) and IS (8,087) in the test (Fig. 3a and g), but Lasso-BLR has the lowest VS$^{0.5}$ (2380); see Fig. 3d. We find that not all combinations produce an improvement in SS$_{RMSE}$ when expanding Fig. 2b to include all model combinations (Fig. 3j), but both Lasso-BLR (2.4%) and EN-BLR (4.2%) do. The computation time required to train both models is under 100 s, but EN-BLR generates 100 predictive scenarios faster (less than 1 s); see Fig. 3m. OMP-SLGPR with $\mathcal{K}_L$ achieved similar results with a much higher computational time of

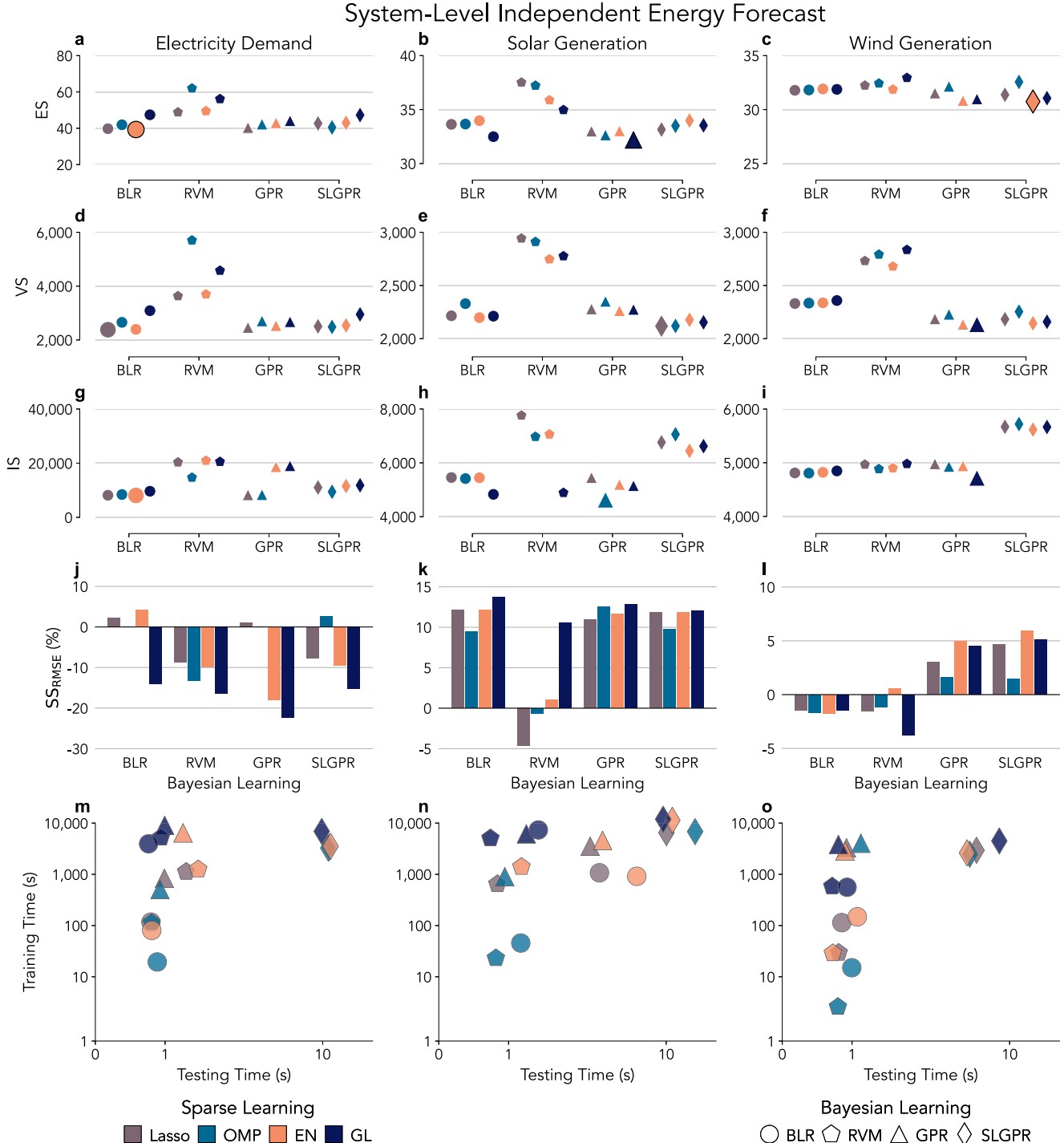

**Fig. 3 | Model selection based on multivariate scoring rules.** Proper scoring rules are applied to evaluate the different day-ahead probability forecasting models of electricity demand (**a**, **d**, **g**), solar generation (**b**, **e**, **h**), and wind generation (**c**, **f**, **i**). The marker color represents Lasso, Orthogonal Matching Pursuit (OMP), Elastic Net (EN), and Group Lasso (GL). The marker shape denotes Bayesian Linear Regression (BLR), Relevance Vector Machine (RVM), Gaussian Process for Regression (GPR), and System-Level GPR (SLGPR)---Multi-Task GPR forecasts a single energy feature but jointly across multiple nodes. The scoring rules are Energy Score (ES) for the electricity demand (**a**), solar generation (**b**), and wind generation (**c**); Variogram Score ($VS^{0.5}$) for the electricity demand (**d**), solar generation (**e**), and wind generation (**f**); and Interval Score (IS) for the electricity demand (**g**), solar generation (**h**), and wind generation (**i**). Lower ES, $VS^{0.5}$, and IS is preferable. The enlarged points in (**a**–**i**) have the lowest values, identifying the best performing model based on that scoring rule. **j**–**l** The Skill Score RMSE-based ($SS_{RMSE}$) compares the proposed forecast with the forecast provided by CAISO (lower is preferable). Training and testing computing time (lower time is preferable) for the proposed day-ahead forecasting models for the electricity demand (**m**), solar (**n**), and wind generation (**o**).

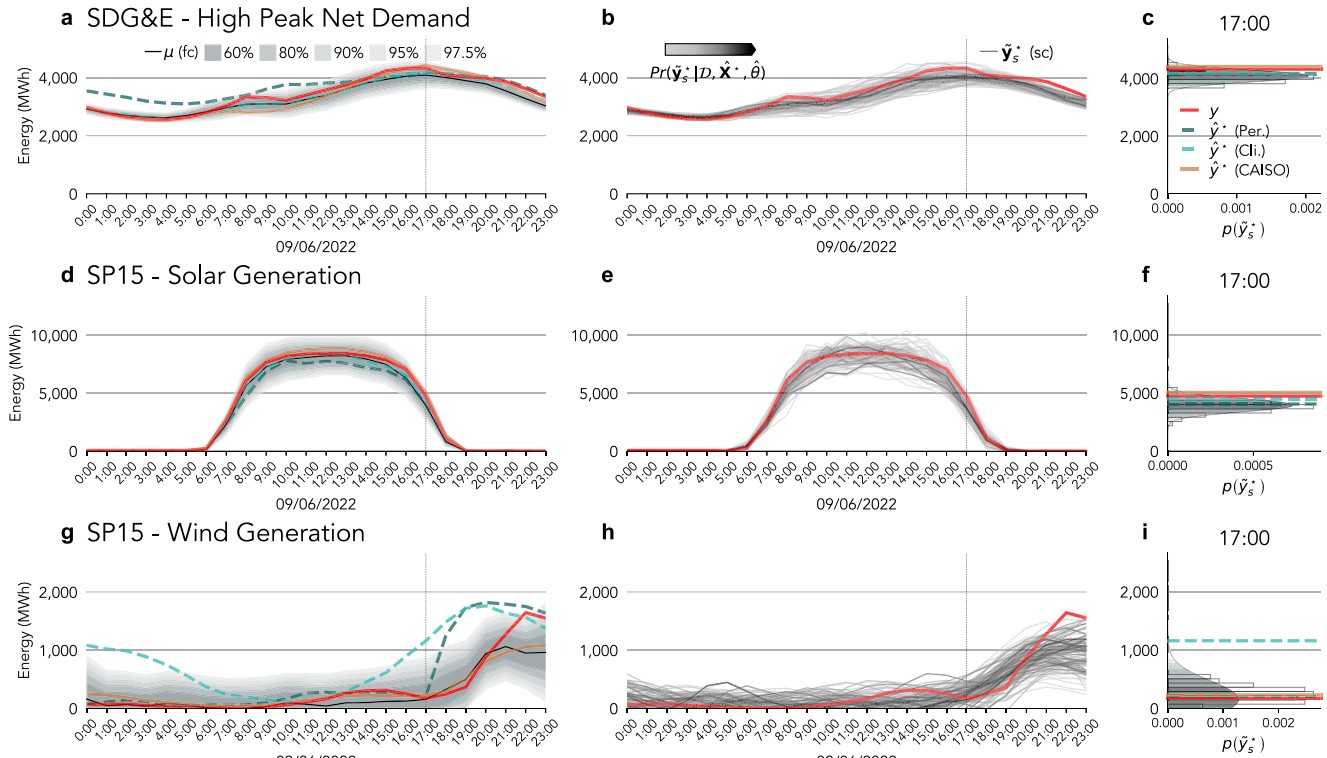

**Fig. 4 | Independent electricity demand, solar and wind generation day-ahead probabilistic forecasts.** Probabilistic day-ahead SDG&E electricity demand forecast during the high peak demand event on Sep 6, 2022, predictive intervals (**a**), predictive scenarios (**b**), and density function detail at 17:00 (**c**). Probabilistic day-ahead solar generation forecast at SP15 in Sep 6, 2022, predictive intervals (**d**), predictive scenarios (**e**), and density function detail at 17:00 (**f**). Probabilistic day-ahead wind generation forecast at SP15 on Sep 6, 2022, predictive intervals (**d**), predictive scenarios (**e**), and density function detail at 17:00 (**f**). **a, d, g** Probabilistic

forecast predictive mean $\boldsymbol{\mu}^\star$ compared to the actual $y^\star$, and the baselines $\widehat{\mathbf{y}}^\star$ (persistence, climatology, and CAISO). The represented predictive intervals (60%, 80%, 90%, 95%, and 97.5%) have a color gradient from dark (60%) to light gray (97.5%). **c, f, i** dashed lines mark detailed hours in (**b, e, h**). The color gradient in the predictive scenarios represents the probability $\Pr(\widetilde{y}_s^\star | \mathcal{D}, \widehat{\Theta}, \widehat{\mathbf{X}}^\star)$ ftive scenarios represents the probabilit of the $s^{th}$ scenario $(\widetilde{y}_s^\star)$ of electricity demand ($\mathcal{L}$), solar ($\mathcal{S}$) or wind ($\mathcal{W}$) generation (darker means higher probability).

around 2000 s (train) and 10 s (test). EN has difficulties identifying the weather features correlated with an electricity demand but selected the discomfort index (Supplementary Note 3) and assigned higher weights around the Bay Area and the Central Valley for NP15 and around the greater Los Angeles for SP16 (Supplementary Fig. 6c–e).

Following the same model selection procedure, GL-GPR with $\mathcal{K}_L$ is the most suitable model for a probabilistic day-ahead solar generation forecast. GL-GPR with $\mathcal{K}_L$ has the lowest ES (32.2) and the highest SS$_{RMSE}$ (13.7%) in Fig. 3b, k Lasso-SLGPR with $\mathcal{K}_L$ has the lowest VS$^{0.5}$ (2,118), and OMP-GPR with $\mathcal{K}_L$ kernel has the lowest IS (4628); see Fig. 3e, h. OMP models consistently have low computation costs (Fig. 3n), and SLGPR models have very low VS$^{0.5}$ but very high IS (Fig. 3e, h). BLR and GPR models achieved similar VS$^{0.5}$, but the OMP-GPR with $\mathcal{K}_L$ had the lowest IS and similar ES (32.6) at lower computation costs (less than 1000 s in training), and it achieved similar SS$_{RMSE}$ (12.5%) than GL-GPR (12.8%). GL selects features across California (Supplementary Fig. 6f–h), but features are consistent and have more weight in regions with installed solar capacity.

The model selected for a probabilistic day-ahead wind generation forecast is EN-SLGPR with a Matérn kernel and parameter $v = 1.5$ ($\mathcal{K}_{M_{1.5}}$). EN-SLGPR with $\mathcal{K}_{M_{1.5}}$ has the lowest ES (30.7) and the highest SS$_{RMSE}$ (5.9%); see Fig. 3c, l. GL-GPR with rational quadratic kernel ($\mathcal{K}_{RQ}$) has the lowest VS$^{0.5}$ (2135) and a SS$_{RMSE}$ of 4.5%; most all GPR and SLGPR models achieved similar results (Fig. 3f and l). GL-GPR also has the lowest IS (4,716), BLR and ARM achieved a similar score (Fig. 3i), but their SS$_{RMSE}$ is negative for most models (Fig. 3l). EN-SLGPR and GL-GPR have similar computing performances in the test ( < 10 s), but their training times are different (around 3000 s and 6000 s), which makes

EN-SLGPR with $\mathcal{K}_{M_{1.5}}$ the most suitable model (Fig. 3o). EN selected features from two regions with high installed wind capacity (Supplementary Fig. 6i, j).

**Predictive density function, intervals, and scenario generation**
To assess the performance of our forecast under stress events, we evaluated our proposed forecast on Sep 2022 (Supplementary Fig. 7). Sep 2022 registered a record high mean temperature in western North America[49], producing several consecutive days of high demand and peak demand[50]. We show the forecasts of the proposed probabilistic day-ahead forecast in southern California (SDG&E and SP15) in Fig. 4, and northern and central California (PG&E, SCE, NP15, and ZP26) in Fig. 4.

The probabilistic day-ahead electricity demand forecast, EN-BLR, estimated SDG&E peak demand in a high peak event on Sep 6, 2022 (Fig. 4a) with less than 5% error. The persistence forecast over-estimated demand during off-peak hours and underestimated demand during peak hours (from 4 pm to 9 pm), while the climatology forecast produced a result similar to the proposed forecast. The CAISO forecast had high-magnitude errors (16%) in the morning but accurately predicted peak demand and time (9 pm). Similarly, the CAISO forecast matches the predictive mean from our forecast but is more accurate at the peak demand hour for the PG&E and SCE (Fig. 4a, d). Our solar generation forecast (GL-GPR with $\mathcal{K}_L$) at SP15 had a lower error than the persistence and CAISO forecast during off-peak hours when the CAISO forecast overestimated the generation and the persistence under-estimated it by 10% (Fig. 4d). In contrast, the climatology forecast produced a result similar to the proposed forecast mean. The

persistence, climatology, and CAISO forecast overestimate solar generation by 10%, every hour, at NP15 and ZP16 (Supplementary Fig. 8g, j). The wind generation forecast (EN-SLGPR with $\mathcal{K}_{M_{1.5}}$) had low-magnitude errors (200 MW) until the evening when the errors reached 600 MW (55%). The persistence forecast produced low errors when the wind generation was negligible (morning and afternoon). The climatology forecast had high errors in the morning and evening. The CAISO forecast was similar to the mean of our forecast. Our forecast at NP15 failed by 50% during peak demand (5 pm) and by 17% during the evening (from 7 pm to 11 pm); see Supplementary Fig. 8m. Similarly, the persistence forecast had low errors, the CAISO forecast had large errors in the morning and evening (50%), and the climatology forecast had large errors.

The probabilistic day-ahead electricity demand forecast (EN-BLR) predicted the peak demand within the 80% interval in the same high peak event for PG&E, SCE, SDG&E (Supplementary Fig. 8a, d, and Fig. 4a). The actuals were outside the 97.5% interval at 10 pm and 11 pm for SDG&E, during off-peak hours for PG&E, and at 9 am and 10 am for SCE. The electricity demand scenarios generated from the predictive density functions capture the time structure (Fig. 4b) and correctly represent the density function at the time of the peak for SDG&E and SCE (Fig. 4c). The actual exceeded the most extreme scenario at night and off-peak hours for PG&E when the forecast error was high (17%); see (Supplementary Fig. 8b).

The uncertainty of the probabilistic day-ahead solar generation forecast (GL-GPR with $\mathcal{K}_L$) at the NP15 (Fig. 4e), SP15 (Supplementary Fig. 8g), and ZP26 (Supplementary Fig. 8j) did not change significantly during off-peak hours despite providing an accurate forecast. In contrast, the forecast does not correctly predict the actual at NP15 from 2 pm to 4 pm but is within the 90% −97.5% interval. The distribution of predictive scenarios approximates the density function at SP15 (Fig. 4e and f) and ZP26 (Supplementary Fig. 8k), and the most extreme scenarios include the actual, even at the time of highest magnitude error (4 pm) at the NP15 (Supplementary Fig. 8h).

The probabilistic day-ahead wind generation forecast (EN-SLGPR with $\mathcal{K}_{M_{1.5}}$) fails from 9 pm with high errors (500 MW), though the actual fell within the 90% interval. The predictive intervals are adaptable but are still wide when the model produces accurate forecasts during the morning (Fig. 4g). The predictive scenarios do not represent the entire range of the density function at the peak demand, and no scenario covers the actual generation at 10 pm (Fig. 4h, i). In contrast, the scenarios in our forecast at NP15 correctly represent the density function, and the scenarios enveloped the actuals (Supplementary Fig. 8m).

## Joint probabilistic day-ahead energy forecast

Joint forecasts can reduce uncertainty and assist ISOs in operating power grids more efficiently by characterizing the dependencies between variables. A Multi-Task Gaussian Process for Regression (MTGPR) model captures the underlying correlation between multiple response variables to generate a joint forecast. However, estimating the joint predictive covariance between response variables with an MTGPR is computationally challenging. The confidence intervals derived from the predictive covariance may not accurately reflect the true distribution of the forecasting errors−i.e., the probability that actual realizations fall outside the upper and lower bounds of the intervals can exceed the stated confidence level (see Section Predictive Density Calibration). The formulation proposed in ref.[45] accurately estimates the joint predictive covariance when the response variables belong to the same energy domain (e.g., electricity demand). However, this limitation persists when the response variables come from different energy domains. To address this, we adopt an approach based on conformal learning[51,52]. In this approach, the dataset is split into training and calibration sets during cross-validation to properly calibrate the confidence intervals in the joint forecast (see Section Experimental Setup).

We assess the joint forecast performance during an stress event when the CAISO forecast had high-magnitude errors on May 2022. The largest curtailment occurred on May 29, as an unusual storm crossed the western United States[53], coincidental with an high CAISO forecast error event (Fig. 5), which produced large VRE curtailments[54]. Similarly, we evaluate the models with the Energy Score (ES), Variogram Score (VS$^{0.5}$), and Interval Score (IS). ES and VS$^{0.5}$ assess distribution and shape of multivariate predictive scenarios ($\widehat{\mathbf{y}}_{t,z}^{\star}$). IS measures how many samples are outside at different confidence intervals (60%, 80%, 90%, 95%, and 97.5%) and for how much. The confidence intervals are derived from the predictive net demand distribution $\mathcal{N}(\widehat{\mu}_t^{\mathcal{N}\star}, \widehat{\sigma}_t^{2\mathcal{N}\star})$ at the system level. The predictive mean of net demand is given by $\widehat{\mu}_{t,z}^{\mathcal{N}\star} = \sum_z \widehat{\mu}_{t,z}^{\mathcal{L}\star} - \widehat{\mu}_{t,z}^{\mathcal{S}\star} - \widehat{\mu}_{t,z}^{\mathcal{W}\star}$ and the variance by $\widehat{\sigma}_t^{2\mathcal{N}\star} = \sum_{z=1}^{Z} \widehat{\sigma}_{t,z}^{2\mathcal{L}\star} + \widehat{\sigma}_{t,z}^{2\mathcal{S}\star} + \widehat{\sigma}_{t,z}^{2\mathcal{W}\star}$. In the case of a joint forecast (SLGPR or NLGPR), the variances are derived from the predictive covariance $\widehat{\sigma}_{t,z}^{2\star} = \text{diag}(\widehat{\mathbf{\Sigma}}_{t,z}^{\star}) + \sigma_{n_{t,z}}^2$ for each energy feature ($\mathcal{L}$, $\mathcal{S}$, and $\mathcal{W}$).

Lasso-NLGPR with a linear kernel $\mathcal{K}_L$ in NP15, SP15, and ZP26 ($\mathcal{K}_L$) had the lowest ES in test (85.3), Lasso-SLGPR with $\mathcal{K}_{L,L,M_{1.5}}$ has the lowest VS$^{0.5}$ (21,528) and EN-BLR the lowest IS (14,009); see Fig. 5a-c, and Supplementary Table 4. The model with higher SS$_{\text{RMSE}}$ is EN-BLR (25.2%), but Lasso-NLGPR has similar score (24.8%); see Fig. 5d. In contrast, SLGPR models have high SS$_{\text{RMSE}}$ but low IS compared to BLR and NLGPR. Lasso-SLGPR and OMP-SLGPR require 5 × more training and testing time than EN-BLR (Fig. 5c). However, the testing time is still less than 60 s. The training time increases from 30 min to greater than 2 h, but the proposed model only requires training updates once every half or one year.

The joint predictive electricity demand, solar, and wind scenarios include the actual demand realization (Fig. 5g) but do not include scenarios with the actual solar generation at 12 pm and 2 pm (Fig. 5j), and the actual wind generation at 9 pm and 11 pm (Fig. 5m); which indicates that 100 predictive scenarios are not sufficient to represent the full range of possible outcomes. Since, during the event of the largest errors between the predictive mean and the actual for the electricity demand (7 pm), solar (2 pm), and wind forecasts (8 pm), the actual electricity demand was within the 90% predictive interval (Fig. 5h), the solar generation was outside the 97.5% predictive interval (Fig. 5k) and the wind generation was within the 80% predictive interval (Fig. 5n). The joint predictive scenario (bright green) represents a mid-low electricity demand (Fig. 5g) with mid-low solar generation from 10 am to 2 pm (Fig. 5j) and mid-low electricity demand with mid-high wind generation (Fig. 5j) from 1 am to 10 am and from 8 pm to midnight. These results are for node NP15, but the findings in node SP15 are similar under high-magnitude forecasting errors (Supplementary Fig. 9).

## Probabilistic day-ahead energy forecast for reserves allocation

Independent system operators (ISOs), including CAISO, procure ancillary services or operating reserves to address day-ahead electricity demand and wind and solar generation forecasting errors[12,55], as well as to prepare for contingency events such as loss of a generator or transmission line[56]; see the estimation of different ancillary services in Section Energy Imbalance. As the growing share of VRE generation adds to the uncertainty introduced by electricity demand, a net demand forecast has become more informative in quantifying reserves requirements[9]. When positive net demand forecast errors (generation overestimation or demand underestimation) exceed upward reserves[57], ISOs import energy from neighboring interconnected regions or shed demand in severe cases. When negative net demand forecast errors (generation underestimation or demand overestimation) exceed downward reserves, ISOs export electricity to

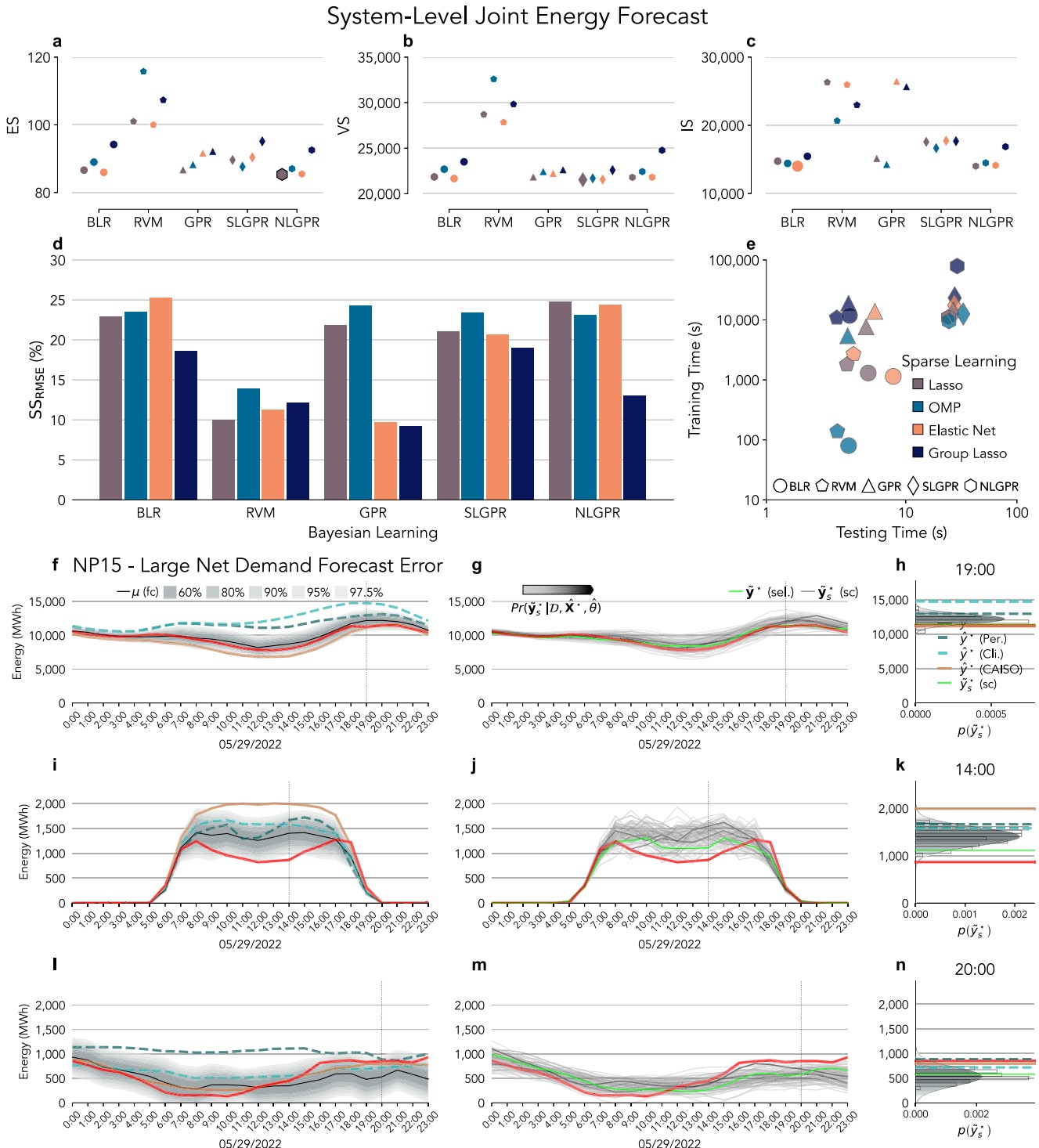

**Fig. 5 | Joint day-ahead energy forecast model selection.** Scores achieved by the proposed forecasting models when evaluated at the system level: **a** Energy Score (ES), **b** Variogram Score (VS$^{0.5}$), **c** Interval Score (IS), **d** Skill Score (SS$_{RMSE}$) compared to CAISO's forecast, and **e** computational time. Lower ES, VS$^{0.5}$, IS, computational time, and higher SS$_{RMSE}$ are preferable. The different colors represent the sparse learning: Lasso, Orthogonal Matching Pursuit (OMP), Elastic Net (EN), and Group Lasso (GL). The shapes represent the Bayesian learning: Bayesian Linear Regression (BLR), Relevance Vector Machine (RVM), Gaussian Process for Regression (GPR), System-Level GPR (SLGPR), and Node-Level GPR (NLGPR). **e** Lasso-NLGPR and EN-

NLGPR have the same computational requirements (hexagons overlap). Lasso-NLGPR forecast during an large CAISO forecast error at NP15 on May 28, 2022. **f**, **i**, **l** Joint forecast predictive mean $\mu^{\star}$, actual $y^{\star}$ and the baseline forecasts $\hat{y}^{\star}$ (persistence, climatology, and CAISO) of PG&E electricity demand (**f**), solar (**i**), and wind generation (**l**) at NP15. The lines are joint scenarios drawn from the predictive density function (the gray color intensity represents their probability). The highlighted scenario is a joint draw of electricity demand (**g**), solar generation (**j**), and wind generation (**m**). Density function details (**h**, **k**, **n**) are of the hours (marked by vertical dashed lines) with the largest error between the predictive mean $\mu_t^{\star}$ and the actual $y_t^{\star}$.

neighboring regions or curtail VRE generation. In California, the CAISO imports from and exports to the Western Energy Imbalance Market[12].

To reduce the risk of forecasting errors, we use the confidence intervals for the net demand forecast that adapt depending on the similarity of the input features to past patterns[22]. This section's experiment validates the hypothesis that using confidence intervals to determine operating reserves will decrease imbalance market trades. The proposed methodology utilizes the predictive density function from the day-ahead net demand forecast to find the confidence interval in which the total aggregated capacity equals the total aggregated capacity allocated following CAISO's methodology as in the Section Energy Imbalance. The area between the selected confidence interval's lower and upper bounds indicates the reserve requirements. This methodology does not determine the aggregated reserves requirement but simply redistributes the aggregated reserves determined by CAISO across time. As such, this methodology is applied only to illustrate the value of using confidence intervals from probabilistic forecasts to determine operating reserves.

The idea is that a day-ahead net demand forecasting model with accurate confidence intervals will allocate the reserves following the proposed methodology more efficiently than CAISO's methodology. Therefore, lower IS in validation is our model selection criterion, since IS is a proper scoring rule that rewards calibrated confidence intervals. We calculate the IS for confidence intervals of 60%, 80%, 90%, 95%, and 97.5% and aggregate to select a model for the application to determine operational reserves. The probabilistic day-ahead net demand forecast with the lowest IS at the system level in the test set is OMP-NLGPR with $\mathcal{K}_L$ (Fig. 6a).

We illustrate the reserves allocation results for Sep 23, 2022, a day in our sample when both CAISO and our method have imbalances. The reserves allocation implemented with CAISO's forecast and method allocates a similar capacity across all day hours. However, the capacity slightly increases during the peak hour and decreases during midday and night hours (Fig. 6c). In contrast, our method of estimating operational reserves based on confidence intervals allocates more reserves during daylight hours (Fig. 6c), displacing reserves from the night to daylight hours. Although mid-day hours experience lower net demand, the uncertainty in the day-ahead solar generation forecast is compared to evening peak or non-solar hours, which drives the higher reserve requirement (Supplementary Fig. 7).

For all days in the test set (243), we evaluate whether the net demand is above the maximum upward reserves where the reserves respond to avoid exports (and VRE curtailment) or imports (and demand shedding) in the imbalance market; see equation (27). For the featured day in (Fig. 6b), using CAISO's reserves allocation methodology, the net demand positive forecast error is greater than the maximum upward reserves available from 8 am to 2 pm, which required energy imports (Fig. 6d). With our proposed reserves allocation methodology, for the same day, the net demand negative forecast error is greater than the maximum downward reserves from 2 am to 6 am, which results in wind generation curtailment (Fig. 6d).

At an aggregate level across all days in test set, the reserve capacity estimated using the CAISO's forecast and reserves allocation method did not produce exports (or VRE curtailment) but required 5 GWh of energy imports per day on average (Fig. 6e). In contrast, the proposed joint forecasting model at the nodal level OMP-NLGPR with $\mathcal{K}_L$, results in exports of 260 MWh and imports of 80 MWh per day on average. The total imbalance, calculated as daily exports (energy curtailment) plus imports (demand shedding), is 340 MWh per day on average. We compare the joint and independent forecasts with the lower IS to verify that a joint energy forecast produces fewer energy imbalances. The forecasting model with independent energy features that achieved the lowest IS, evaluated on the days in the test set, is EN-BLR (Supplementary Fig. 10a). This model produces more imbalances (410 MWh per day) than the joint energy features (OMP-NLGPR)

model. On average, it has fewer exports (110 MWh) but more imports (300 MWh per day).

The probabilistic day-ahead net demand forecast with the lowest total imbalances is an ensemble model formed by the joint energy forecasting models with the lowest IS in each node: OMP-NLGPR with $\mathcal{K}_L$ at NP15 (Supplementary Fig. 10i), Lasso-NLGPR with $\mathcal{K}_L$ at SP15 (Supplementary Fig. 10j), GL-NLGPR with $\mathcal{K}_L$ at ZP26 (Supplementary Fig. 10k). The daily average exports (VRE curtailment) are 160 MWh, and imports (demand shedding) are 180 MWh, producing the lowest imbalance (340 MWh). The independent ensemble day-ahead net demand forecast with the lowest IS for each energy feature has EN-BLR for electricity demand (Supplementary Fig. 10f), EN-BLR for solar (Supplementary Fig. 10g), and EN-GPR with $\mathcal{K}_L$ for wind generation (Supplementary Fig. 10h). The average daily energy exports (or VRE curtailment) are 120 MWh, and imports are 230 MWh, while the average imbalance is 350 MWh. This result demonstrates that the IS is an effective indicator for selecting a reserve allocation method based on confidence intervals. Ensemble probabilistic day-ahead energy forecasts lead to less total imbalance in power grid operation in line with the No Free Lunch theorem[58].

Additionally, jointly forecasting energy features minimizes total imbalance and reduces the bias, further enhancing the efficiency of the reserve allocation method and potentially minimizing the operating costs, while not affecting or even improving the effectiveness of the net demand forecast for coordinating the wholesale electricity markets and, overall, reducing the risk of large forecasting errors.

## Discussion

Our results, using the case study of the California Independent System Operator (CAISO), show that a probabilistic day-ahead forecast for wind, solar, and electricity demand performs better than deterministic forecasts (Fig. 2) with the additional benefit of having a predictive density distribution to generate predictive scenarios and derive intervals. Consequently, probabilistic day-ahead energy forecasts based on Bayesian learning are more suitable and versatile for applications in power systems, including resource assessment, stochastic operational planning, and operational reserve allocation (Fig. 6).

We analyze our results to isolate the impacts on the performance of sparse learning, the joint distribution, and the kernel function. We first assess the sparse learning impact on the performance by adjusting the hyperparameters to achieve different sparsity levels (ranging from 10 to 1000). We find that the best models consistently include features in the magnitude order of 100s for electric load and solar and 10s for wind (Supplementary Fig. 6). However, sparse learning is always necessary to reduce the computational complexity of the problem. The Relevance Vector Machine (RVM) is an indicator of when the sparse model is unnecessary. The RVM underperforms because it cannot identify the most relevant features (Fig. 2b, e).

We then validate whether modeling the joint distribution across nodes or features improves performance. The performance increased when incorporating the joint distribution (across energy features at the nodal level) into the Gaussian Process for Regression (NLGPR) for electricity demand with Lasso, and solar generation with Elastic Net (EN), compared to independent models (Fig. 2b, c). We can also confirm this in ZP26, which does not have wind generators (Fig. 2e, f). Additionally, we assess the joint distribution (across nodes for a single energy feature) at the system level (SLGPR) when each feature has a kernel, and find that it provides an advantage for wind generation (Fig. 5l).

Finally, we can isolate the effect of the kernel function on the performance. The performance of wind generation declines because electricity load and solar generation favor a linear kernel ($\mathcal{K}_L$), which negatively impacts the performance when evaluating net demand at the nodal level (Fig. 2f). Wind generation requires an EN-NLGPR with rational quadratic ($\mathcal{K}_{RQ}$) or Matérn ($\mathcal{K}_{M_{1.5}}$) kernel (Supplementary

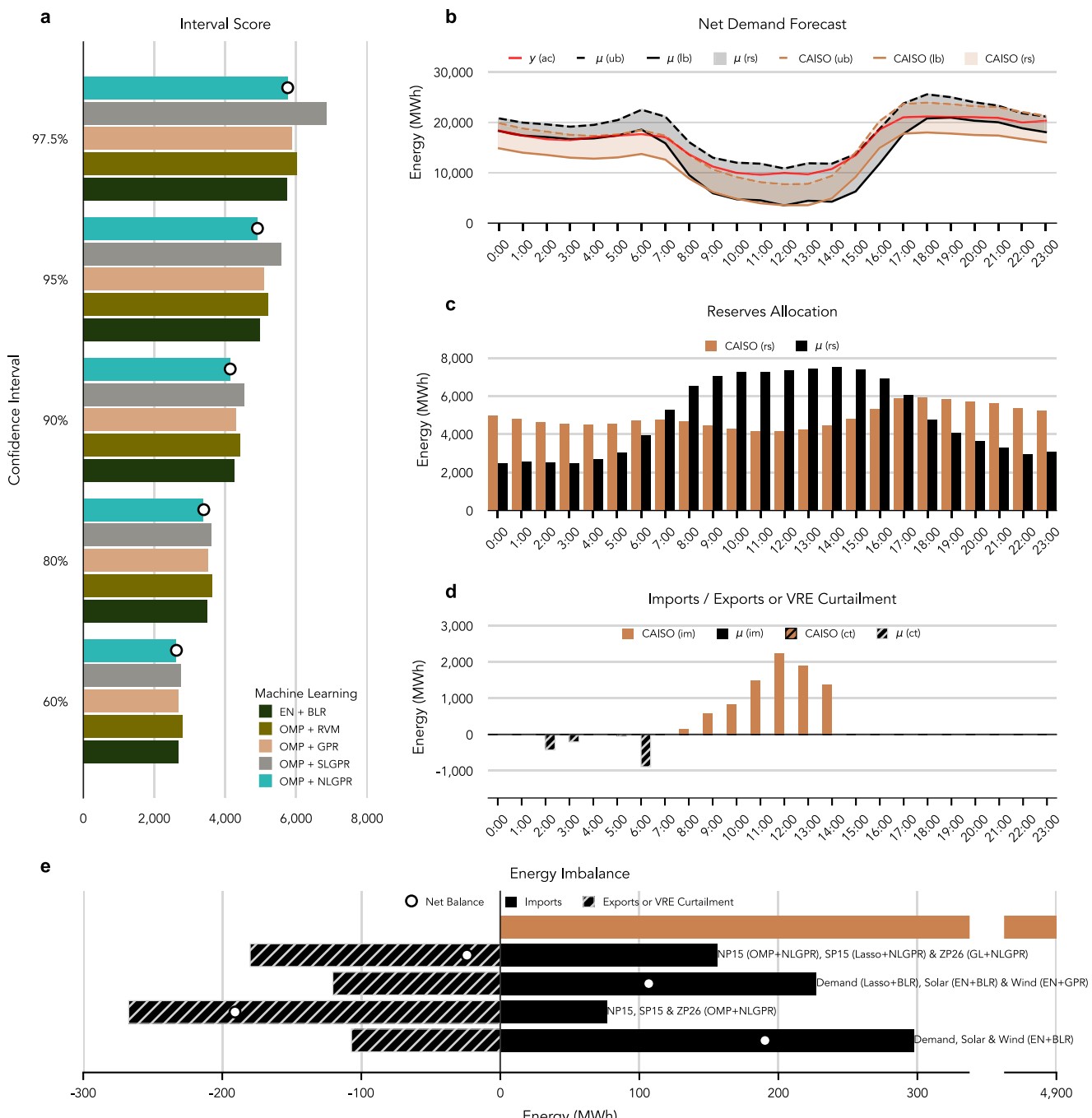

**Fig. 6 | Day-Ahead operational reserves allocation. a** Interval Score (IS) from the proposed independent and joint probabilistic day-ahead forecasts calculated for the 60%, 80%, 90%, 95%, and 97.5% confidence intervals (lower values are better). The white dots indicated the best model. The independent and joint forecasts are aggregated at the system level. The independent models are Elastic Net and Bayesian Linear Regression (EN-BLR), Orthogonal Matching Pursuit (OMP) and Relevance Vector Machine (OMP-RVM), and OMP and Gaussian Process for Regression (GPR). The joint models are OMP and System-Level GPR (SLGPR), as well as OMP and Node-Level GPR (NLGPR). **b** Actual net demand (**y**) and net demand forecasts for OMP-NLGPR (**μ**) and CAISO on Sep 22, 2022. Net demand forecast reserves' upper bound (*ub*) and lower bound (*lb*) shown by lower solid and upper dashed lines and reserves (*rs*) shown by shaded areas between solid and dashed lines---the reserves' upper and lower bounds are the confidence interval lower and upper bounds for OMP-NLGPR (**μ**). **c** Upward + downward reserves allocation for CAISO and OMP-NLGPR (**μ**) forecasts. **d** Energy imports (*im*) (and demand shedding if energy is not available in the regional energy imbalance market) and exports (and wind or solar curtailment, *ct*, if demand is lacking in the energy imbalance market) resulting from the CAISO and the proposed allocation methodology using OMP-NLGPR (**μ**) forecast (positive means energy imports required, while negative means energy exports available). **e** Average imports and energy curtailment obtained from CAISO's forecast and the proposed ensemble of probabilistic day-ahead net energy forecasts across the simulated period from May 2022 to Feb 2023 (test samples). White dots represent the net balance between imports, exports, and energy curtailment.

Tables 1–4), indicating the existence of non-linear relations between the input features and wind generation. Multiple kernel learning exploits non-linear relations by explicitly defining a transformation for each feature source[59], but the current MTGPR library does not support multiple kernel functions[45].

As a consequence, Elastic Net with Bayesian Linear Regression (EN-BLR) performs better overall. Still, when we consider ensembles to take advantage of the joint forecasts and evaluate them with proper scoring rules, we observe that joint forecasts are advantageous in power system applications, such as reserve allocations (Fig. 6e).

According to previous research, the most suitable model for a probabilistic day-ahead electricity demand forecast was the RVM[60]. In contrast, our research shows that Bayesian methods based on kernel learning improve the performance of an RVM by exploiting underlying non-linear relations in the input feature space to produce a joint forecast across features and nodes (Fig. 5). A previous non-linear probabilistic forecast of wind generation proposed a Radial Basis Function kernel ($\mathcal{K}_{RBF}$). Instead, after cross-validating different kernels (including $\mathcal{K}_{RBF}$), we found that the Matérn kernel ($\mathcal{K}_{M_{1.5}}$) considerably reduces the forecasting errors in our experiments[61].

Other recent studies have introduced day-ahead forecasting methods that minimize the energy score[62]. Yet our findings suggest that selecting a model based solely on a score leads to poor performance in other scores (skill score and interval score) under certain conditions (Fig. 5). Bayesian learning models that optimize a purely probabilistic loss and provide probability calibration frameworks, in conjunction with examining the model performance across multiple proper scoring rules, are a more explainable and risk-averse approach to encourage industry adoption. Furthermore, the best-performing models consistently selected sparse representations of the input space to enhance the effectiveness of a probabilistic day-ahead forecast (Supplementary Fig. 6), contradicting earlier approaches that relied on dense weather features from NWFs[35].

For California, our findings indicate more uncertainty in electricity demand during daylight hours compared to night hours. This increase in the electricity demand uncertainty during daylight hours could likely be explained by social or behavioral patterns, rather than solely by weather features. Additional research is needed to understand which social patterns (e.g., traffic, rain, or electric vehicle charging patterns) significantly impact electricity demand forecasts and how to quantify these effectively. Contrarily, the confidence interval during the daytime or nighttime hours remains conservative even when baseline models and the proposed probabilistic forecast align with actual outcomes in solar (Fig. 4d) and wind generation (Fig. 4g), which reflects uncertainty introduced by the predicted weather features from the HRRR, congestion or generator outages.

CAISO's approach for allocating operational reserves combines the deterministic day-ahead electricity demand forecast with historical forecasting errors[55]. In contrast, our approach utilizes the net demand predictive interval equivalent to the total operational reserves allocated following CAISO's approach (Fig. 6b). The bias in CAISO's forecast reduces curtailment but overestimates solar generation, producing substantial energy imports (Fig. 6e). This bias allows CAISO to effectively predict peak capacity and ramp-up steepness, which are critical during stress events on the power grid. However, during hours of high net demand, CAISO's forecast performs worse than persistence and climatology, leading to high energy imports from the energy imbalance market (Fig. 2j). The proposed joint probabilistic day-ahead forecast has a negligible bias, making it more suitable for determining the reserves (Fig. 6e). Furthermore, optimizing the lower and upper predictive intervals could reduce energy curtailment.

The proposed dynamic reserves allocation serves as an example of how to utilize a probabilistic forecast in power system operations. Our imbalance market simulations do not consider real-time prices. In addition, the simulations do not enforce the minimum stable capacity constraint on generators during unit commitment. We also do not include the largest committed unit capacity and contracted imports and exports in CAISO's method for reserves allocation, since this information was not available for comparison[56].

We intentionally integrated the ability to generate predictive scenarios associated with probabilities in a day-ahead forecast to assess the Conditional Value at Risk (CVaR)—a risk measure to estimate the expected losses in the tails of a distribution—for dynamic reserves allocation and analyze the impact of spatial correlations on both demand and generation on grid congestion, considering congestion and committed units to account for energy imports and exports more accurately. However, this investigation will require methods and datasets beyond the scope of this study.

The results may vary when applied to other ISOs due to different regional demand patterns, generation mixes, and specific operational requirements (i.e., forecasting horizon and lag). However, the proposed joint day-ahead probabilistic forecasting method remains applicable and transferable with the appropriate datasets. Since our net demand forecast is based on a top-down approach that learns short-term patterns aggregated at the three main zones in CAISO, we implicitly model the bottom-up breakdown of disaggregated demand (e.g., residential, industrial, and weather-sensitive loads) and distributed generation. While this abstraction level may be insufficient for long-term disaggregated bottom-up demand forecasts, it is effective in our aggregated day-ahead net demand forecast application.

In summary, the proposed probabilistic day-ahead forecasts, based on Bayesian and kernel learning, improve the accuracy and effectively capture the uncertainty in electricity demand, wind, and solar generation. By jointly forecasting these energy features, the model enhances reserve allocation and generates electricity demand and supply scenarios for risk assessment in power grid planning. Ultimately, the proposed reserve allocation method has the potential to reduce energy imports and curtailment, promoting the integration of variable renewable energy sources and enabling a more efficient, sustainable, and resilient electricity system.

## Methods

This section includes the description of the Data, the Processing and Filtering, the Theoretical Background, the Data Structure, and the Experimental Setup. The Theoretical Background is divided into Sparse Learning and Bayesian learning, where the different methods are explained. The sections about Scenario Smoothing and Predictive Density Calibration propose approaches to overcome limitations.

### Data

The weather forecast is from the HRRR (Supplementary Note 5). It has a spatial resolution of 3 × 3 km and covers the Continental United States (CONUS) and Alaska. The NWF assimilates radar information every 15 min over 1 h to add further detail to the RAP weather forecast, which has a spatial resolution of 13 × 13 km. HRRR provides 192 weather variables, which include actuals (i.e. analyzed) and forecasts at different altitudes in the atmosphere. The forecast has a 48 h horizon and is provided 4 times a day (00, 06, 12, and 18 UTC). This investigation assesses the following weather variables obtained from the HRRR: atmospheric pressure, dew point, relative humidity, temperature, direct long-wave radiation flux, direct short-wave radiation flux, and wind velocity components at 10 m and 80 m.

The wind velocity magnitude is interpolated by applying the Power Law in supplementary equation 8 at 60 m, 100 m, and 120 m from the wind velocity at 10 m and 80 m[63]. The discomfort index is derived from the relative humidity and the temperature[64], see supplementary equation 9. The Global Horizontal Irradiance (GHI) is estimated with a theoretical model[65]. The theoretical model requires the elevation and latitude of each point in the 3 × 3 km, in addition to

the date (i.e., year, month, day, and hour). Ultimately, the model is capable of providing intra-hour estimation. The elevation information is from the Global Multi-resolution Terrain Elevation Data 2010 (GMTED2010)[66], developed by the U.S. Geological Survey (USGS) and the National Geospatial-Intelligence Agency (NGA). The elevation estimation comprises data assimilated from multiple sources (i.e., radar and satellite information).

The aggregated electricity demand and solar and wind generation are from the Open Access Same-time Information System (OASIS) platform maintained by CAISO[67]. This platform provides real-time information about the transmission system and electricity market operation within the Western Interconnection[68]. OASIS publishes DA operational forecasts and actual realizations with a 1 h resolution for each training hub and utility in CAISO. Solar generation is available in NP15 (north), ZP26 (central), and SP15 (south) trading hubs. Wind generation is available in NP15 and SP15 trading hubs. Electricity demand forecasts and actuals are also available for MWD, PG&E (north-central), SCE (south-central), and SDG&E (south) utility companies. MWD and VEA are water companies and mainly operate hydroelectric pumps and storage systems, so we do not consider them in this investigation.

The solar and wind power plant locations and specifications are part of the California Energy Commission (CEC) critical infrastructure geospatial datasets[69]. The dataset contains information about the energy type and the total nameplate capacity of all plants with a nameplate capacity ≥1MW in California. It is updated annually from the CEC QFER-1304 Power Plant Owner Reporting Database. This investigation uses solar and wind plant locations in Supplementary Fig. 2b.

The information about the population density in California is from the Gridded Population of the World, Version 4 (GPWv4) dataset (see Supplementary Fig. 2b). GPWv4 provides high spatial resolution worldwide sub-national population density for multiple years in the number of persons per km$^2$, with counts consistent with national census and population register[70]. This investigation utilizes data from the 2020 update.

### Processing and filtering

The multiple spatial data sources (i.e., HRRR, GPWv4, and GMTED2010) were interpolated to have the same spatial resolution. The original resolution of HRRR is approximately $1.7' \times 1.7'$, GPWv4 is $2.5' \times 2.5'$, and GMTED2010 is $30'' \times 30''$. The final resolution is $7' \times 7'$ equivalent to approximately $13 \times 13$ km grid. The method implemented was 2-dimensional nearest-neighbor interpolation. The result is a spatial forecast in a $104 \times 88$ grid ($N \times M$) per weather feature.

The HRRR provides reanalyzed observations and weather forecasts. We downloaded the reanalyzed weather features that match the energy feature time series: electricity demand ($\mathcal{L}$), solar ($\mathcal{S}$), and wind generation ($\mathcal{W}$). We call them the reanalysis dataset ($\mathcal{A}$). Similarly, we downloaded the matching HRRR forecast from the previous day at 00 UTC (4 pm PTZ) for the same energy feature time series, and we call it the forecast dataset ($\mathcal{F}$). We define a sample in the time series $k \in (1, \infty]$ from the reanalysis or the forecast datasets as having a 1-day resolution and containing information for each hour $t$ in a day $T = 24$. In particular, the sample $k$ for hour $t$ is $\mathbf{X}_{k,t} \in \mathbb{R}^{N \times M}$ when is from the reanalysis dataset ($\mathcal{A}$), and $\hat{\mathbf{X}}_{k,t} \in \mathbb{R}^{N \times M}$ when is from the forecast dataset ($\mathcal{F}$).

The second part of the processing is to remove spatial dimensions from the HRRR forecast that do not contain population, or solar or wind power plants in proximity. However, the spatial distribution of errors in the weather forecast is unknown; we include information from nearby regions to the plants and let the sparse learning method select the most suitable features. We calculate the population density to define the electricity demand mask $\psi^{\mathcal{L}}$, the solar capacity density for solar mask $\psi^{\mathcal{S}}$, and the wind generation density for wind mask $\psi^{\mathcal{W}}$. The

solar and wind density is derived from the power plant locations in Supplementary Fig. 2b. We apply the masks to the low-resolution weather features from the HRRR forecast. A detailed explanation of the spatial filtering steps is in Supplementary Note 4.

### Theoretical background

The canonical formulation of a multivariate regression problem, having a set of $N$ observations $\mathcal{D} = \{(y_i, \mathbf{x}_i)\}$, is

$$y_i = \mathbf{w}^\top \mathbf{x}_i + \varepsilon_i, \tag{1}$$

where the response variable is a scalar $y_i \in \mathbb{R}$, and the covariates are feature vectors $\mathbf{x}_i \in \mathbb{R}^D$. The error term $\varepsilon_i \sim \mathcal{N}(0, \sigma_n^2)$ is assumed a i.i.d. random variable, and $\mathcal{N}(\cdot)$ is a Normal distribution.

### Sparse learning

The objective of the sparse learning is to discover which weather variables are more informative about the electricity demand ($y_{k,t,z}^{\mathcal{L}}$) and solar ($y_{k,t,z}^{\mathcal{S}}$) and wind ($y_{k,t,z}^{\mathcal{W}}$) generation time series. Note that $k = 1, \ldots, K$ represents the day, and $t = 1, \ldots, T$ is the hour. Sparse learning models aim to find a solution to equation (1) in which only fractions of the model parameters $\mathbf{w}$ are non-zero. This part is the first stage of the workflow (Supplementary Fig. 2a) and utilizes reanalyzed weather features from the HRRR, $\mathcal{A} = \{(y_{j,z}, \mathbf{x}_j) | \forall j = 1, \ldots, K \cdot T, \forall z, = 1, \ldots, Z\}$, where $\mathbf{x}_j \in \mathbb{R}^{D_1 \times 1}$ (as found in the Feature Vectors for Sparse Learning) and $z$ represent each node (NP15, SP16, and ZP26).

**Lasso.** This model was introduced for geophysical problems with high-dimensional data to perform variable selection (i.e., bandwidths) while providing interpretability[71]. The objective is to reduce the complexity of a linear regression model by selecting a reduced number of covariates. The Lagrangian formulation of Lasso is,

$$\hat{\mathbf{w}}_z = \underset{\mathbf{w}_z}{\arg\min} \left\| \mathbf{y}_z - \mathbf{w}_z^\top \mathbf{X} \right\|_2^2 + \lambda \left\| \mathbf{w}_z \right\|_1, \tag{2}$$

where $\| \cdot \|_1$ and $\| \cdot \|_2$ represent the L$_1$-norm and L$_2$-norm, respectively, and $\lambda$ is the regularization term[72].

**Orthogonal matching pursuit (OMP).** It is the orthogonal version of Matching Pursuit[73]. The primal formulation of OMP is similar to Lasso. The difference is that OMP implements the L$_0$-norm of the parameters $\mathbf{w}_z$ instead of the L$_1$-norm[74],

$$\hat{\mathbf{w}}_z = \underset{\mathbf{w}_z}{\arg\min} \left\| \mathbf{y}_z - \mathbf{w}_z^\top \mathbf{X} \right\|_2^2, \text{ s.t. } \left\| \mathbf{w}_z \right\|_0 \leq \beta. \tag{3}$$

The hyperparameter $\beta$ represents the maximum number of non-zero elements in the model, and $\| \cdot \|_0$ is the L$_0$-norm.

**Elastic net.** This model adds a quadratic regularization term (L$_2$-norm) to the Lasso formulation to overcome the potential saturation (selecting many variables) or group (selecting a unique variable in a group) selection problems[75]. The Elastic Net formulation is

$$\hat{\mathbf{w}}_z = \underset{\mathbf{w}_z}{\arg\min} \left\| \mathbf{y}_z - \mathbf{w}_z^\top \mathbf{X} \right\|_2^2 + \Omega_1 \left\| \mathbf{w}_z \right\|_1 + \Omega_2 \left\| \mathbf{w}_z \right\|_2^2, \tag{4}$$

the hyperparameters $\Omega_1$ and $\Omega_2$ weight the regularization terms. If $\Omega_1 = 0$ or $\Omega_2 = 0$, the model is equivalent to Ridge Regression or the Lasso, respectively.

**Group Lasso.** This model is an extension of the Lasso with grouped covariates[76]. We apply the Lasso regularization (L$_1$-norm) to all model coefficients, and the group regularization (L$_1$-norm) to the coefficients

grouped by weather features in a location (i.e., coordinate pairs). The $L_2$-norm group regularization is not squared, which makes the penalty non-differentiable at zero, enabling the group variable selection. Its optimization problem is

$$\hat{\mathbf{w}}_z = \underset{\mathbf{w}_z}{\mathrm{argmin}} \left\| \mathbf{y}_z - \left( \sum_{c \in \mathcal{C}} \mathbf{w}_{z,c}^{\top} \mathbf{X}_c \right) \right\|_2^2 + \xi_1 \|\mathbf{w}_z\|_1 + \xi_2 \sum_{c \in \mathcal{C}} \sqrt{d_c} \|\mathbf{w}_{z,c}\|_2, \quad (5)$$

$\mathbb{R}^{d_c \times 1}$ are the model parameters for group $c$, $d_c$ is the number of dimensions in group $c$, and $\mathcal{C}$ is the total number of groups. $\xi_1$ is the Lasso regularization and $\xi_2$ is the group regularization hyperparameter. If $\xi_2 = 0$ the model becomes equivalent to the Lasso. The optimal coefficients are found with the fast iterative shrinkage-thresholding algorithm[77].

## Bayesian learning

Bayesian learning models estimate the uncertainty in the prediction produced by the intrinsic epistemic uncertainty in the parameters and aleatory uncertainty in the observations (i.e., noise). This part is the second stage of the workflow (Supplementary Fig. 2a) and utilizes the forecasted weather features from the HRRR, $\mathcal{F} = \{(y_{k,t,z}, \hat{\mathbf{x}}_{k,t}) | \forall k = 1, \ldots, K, \forall t = 1, \ldots, T, \forall z = 1, \ldots, Z\}$, where $\hat{\mathbf{x}}_{k,t} \in \mathbb{R}^{D_2 \times 1}$ (as found in the Pattern Vectors for Bayesian Learning). Index $z$ corresponding to the node number is omitted from the nomenclature for simplicity.

**Bayesian linear regression (BLR).** The objective in this model is to find the parameters $\mathbf{w}$ that maximize the posterior probability[72],

$$p\left(\mathbf{w}|\hat{\mathbf{X}}, \mathbf{y}, \sigma_n^2\right) \propto p\left(\mathbf{y}|\hat{\mathbf{X}}, \mathbf{w}, \sigma_n^2\right) p\left(\mathbf{w}|\boldsymbol{\Sigma}_p\right). \quad (6)$$

The distribution of the response variable is $p(\mathbf{y}|\hat{\mathbf{X}}, \mathbf{w}, \sigma_n^2) \sim \mathcal{N}(\mathbf{w}^{\top}\hat{\mathbf{X}}, \sigma_n^2)$, and the prior distribution of the model parameters is $p(\mathbf{w}|\boldsymbol{\Sigma}_p) \sim \mathcal{N}(\mathbf{0}, \boldsymbol{\Sigma}_p)$, where $\boldsymbol{\Sigma}_p = \sigma_p^2 \mathbf{I}_{D_2 \times D_2}$. In addition, it is possible to regularize the model by adding a conjugate prior of the model hyperparameters $\hat{\theta} = \{\sigma_n^2, \sigma_p^2\}$, so that $p(\sigma_n^2|\alpha_n, \beta_n) \sim \mathcal{G}(\alpha_n, \beta_n)$ and $p(\sigma_p^2|\alpha_p, \beta_p) \sim \mathcal{G}(\alpha_p, \beta_p)$, where $\mathcal{G}(\cdot)$ is a gamma distribution. This is known as Bayesian hierarchical linear regression (Supplementary Fig. 3a). The optimal parameters in hierarchical prior $\alpha_n, \beta_n, \alpha_p$ and $\beta_p$ are found with a Gaussian approximation[78].

**Relevance vector machine (RVM).** The prior of $\mathbf{w}$ is different when implementing the Automatic Relevance Determination (ARD) mechanism in BLR[79,80]. Here, the prior $\mathbf{w} \sim \mathcal{N}(\mathbf{0}, \boldsymbol{\Gamma})$, has a standard deviation $\gamma_j$ for each parameter $w_j$, defined as $\boldsymbol{\Gamma} = \mathrm{diag}([\gamma_1 \cdots \gamma_{D_2}])$. This model has a hyperparameter $\gamma_0$ which defines the threshold to eliminate dimensions in the input space[81]; see Supplementary Fig. 3b.

**Gaussian process for regression (GPR).** The kernel trick enables a linear model to have nonlinear properties. A kernel function is a positive definite function that maps a feature vector $\mathcal{K}(\cdot, \cdot) : \hat{\mathcal{X}} \times \hat{\mathcal{X}} \to \mathcal{R}$ into reproducing kernel Hilbert space $\mathcal{H}$ spanned by a function $\varphi : \hat{\mathcal{X}} \to \mathcal{H}$ and reproduced by the inner product $\mathcal{K}(\hat{\mathbf{x}}_i, \hat{\mathbf{x}}_i') \triangleq \langle \varphi(\hat{\mathbf{x}}_i), \varphi(\hat{\mathbf{x}}_i') \rangle$. As a consequence, applying the Generalized Representer Theorem to the model parameters allows us to express them as a linear combination of the data $\mathbf{w} = \boldsymbol{\Sigma}_p^{1/2} \boldsymbol{\Phi} \boldsymbol{\alpha}$. In the context of BLR, the extension is called GPR[46]. The Maximum A Posteriori (MAP) estimation of $\boldsymbol{\alpha}$ is

$$p\left(\boldsymbol{\alpha}|\boldsymbol{\Phi}, \mathbf{y}, \theta, \sigma_n^2\right) \propto p(\mathbf{y}|\boldsymbol{\Phi}, \boldsymbol{\alpha}, \theta, \sigma_n^2) p\left(\boldsymbol{\alpha}|\boldsymbol{\Sigma}_p, \theta\right), \quad (7)$$

the likelihood is $p(\mathbf{y}|\boldsymbol{\Phi}, \boldsymbol{\alpha}, \theta, \sigma_n^2) \sim \mathcal{N}(\boldsymbol{\alpha}^{\top} \boldsymbol{\Phi}^{\top} \boldsymbol{\Sigma}_p \boldsymbol{\Phi}, \sigma_n^2)$ and the prior is $p(\boldsymbol{\alpha}|\boldsymbol{\Sigma}_p, \theta) \sim \mathcal{N}(\mathbf{0}, \boldsymbol{\Phi}^{\top} \boldsymbol{\Sigma}_p \boldsymbol{\Phi})$. Similarly, the optimal representation of the dual parameters $\hat{\boldsymbol{\alpha}}$ has analytical solutions, and the optimal

hyperparameters $\hat{\theta}$ are found by minimizing the Negative Marginal Log-Likelihood (NMLL). The plate diagram is in Supplementary Fig. 3c.

**Multi-task Gaussian process for regression (MTGPR).** The proposed multi-tasks regression problem[82,83] aims to estimate $\tau$ response variables $\mathbf{y}_k \in \mathbb{R}^{\tau}$, in vector form $\mathbf{y}_k = [y_{k,1} \cdots y_{k,\tau}]$, from a feature vector $\hat{\mathbf{x}}_k$ mapped $\varphi(\hat{\mathbf{x}}_k)$ into a reproducing kernel Hilbert space $\mathcal{H}$ endowed with a dot product $k(\hat{\mathbf{x}}_k, \hat{\mathbf{x}}_l)$, where function $k(\cdot, \cdot)$ is a Mercer's kernel[84], with the following canonical model,

$$\mathbf{y}_k = \mathbf{W}^{\top} \varphi(\hat{\mathbf{x}}_k) + \boldsymbol{\varepsilon}_k, \quad (8)$$

where $\mathbf{W}$ is the matrix of primal parameters $\mathbf{W} = [\mathbf{w}_1 \cdots \mathbf{w}_{\tau}]$ that have a dual representation $\mathbf{W}^{\top} = \mathbf{A} \boldsymbol{\Phi}^{\top}$. The error term $\boldsymbol{\varepsilon}_k$ in the multi-task regression problem is a vector $\boldsymbol{\varepsilon}_k = [\varepsilon_{k,1} \cdots \varepsilon_{k,\tau}]$ that we assumed to have Gaussian distribution $p(\boldsymbol{\varepsilon}_k) \sim \mathcal{N}(\mathbf{0}, \boldsymbol{\Sigma}_n)$ with zero mean and covariance matrix $\boldsymbol{\Sigma}_n$. Under this assumption, the likelihood function is also Gaussian $p(\mathbf{y}_k|\mathbf{W}, \boldsymbol{\Phi}, \boldsymbol{\Sigma}_n) \sim \mathcal{N}(\mathbf{W}^{\top} \varphi(\hat{\mathbf{x}}_k), \boldsymbol{\Sigma}_n)$, and we can obtain the MAP estimation of the dual parameters $\mathbf{A}$ by assuming a Gaussian prior on the prior parameters $p(\mathrm{vec}(\mathbf{W})) \sim \mathcal{N}(\mathbf{0}, \mathbf{C} \otimes \boldsymbol{\Sigma}_p)$, where $\mathbf{C}$ is the inter-task covariance, $\boldsymbol{\Sigma}_p$ is the parameters covariance and $\otimes$ denotes the Kronecker product between matrices.

In the problem at hand, we explore $\tau$ as the number of nodes (NP15, SP15, ZP26), so $\tau = 3$ when forecasting electricity demand ($\mathcal{L}$) and solar generation ($\mathcal{S}$). When forecasting wind ($\mathcal{W}$), $\tau = 2$, since wind is only available in the NP15 and SP15 nodes. Similarly, when forecasting all energy features ($\mathcal{L}, \mathcal{S}$ and $\mathcal{W}$) $\tau = 3$ in node NP15 and SP15, but node ZP26 only has electricity demand and solar generation ($\tau = 2$).

In a conditional one-output likelihood multi-task Gaussian process for regression (Cool-MTGPR)[45], the task of estimating $\tau$ regressors $\mathbf{y}_k \in \mathbb{R}^{\tau}$ from predictor $\varphi(\hat{\mathbf{x}}_k)$ is done with the model

$$y_{k,\tau} = \varphi(\hat{\mathbf{x}}_k)^{\top} \mathbf{w}_{\tau} + \varepsilon_{k,\tau} = \varphi(\hat{\mathbf{x}}_k)^{\top} \mathbf{w}_{x,\tau} + \mathbf{y}_{k,1:\tau-1}^{\top} \mathbf{w}_{y,\tau} + \varepsilon_{k,\tau}. \quad (9)$$

In this formulation, each factorized task $\mathbf{y}_k$ is modeled as dependent of the previous ones, and therefore the corresponding weights are split into $\mathbf{w}_{k,\tau} \in \mathcal{H}$ for the input sample $\hat{\mathbf{x}}_k$ and $\mathbf{w}y, \tau$ for the previous tasks. Indeed, weight vectors $\mathbf{w}_{\tau}$ in an MTGPR can be recovered as $\mathbf{w}_{\tau} = \mathbf{w}_{x,\tau} + \mathbf{W}_{1:\tau-1} \mathbf{w}_{y,\tau}$. Here, model error $\boldsymbol{\varepsilon}$ is assumed to have the form of a Gaussian distribution $p(\boldsymbol{\varepsilon}_k) \sim \mathcal{N}(\mathbf{0}, \boldsymbol{\Sigma}_p)$.

By applying the chain rule of probability to the standard joint multitask likelihood, we can factorize it into a product of conditional probabilities, each one corresponding to each one of the conditional tasks in equation (9),

$$p(\mathrm{vect}(\mathbf{Y})|\boldsymbol{\Phi}, \mathbf{W}) = \prod_{k=1}^{T} p\left(\mathbf{y}_k|\mathbf{Y}_{1:k-1}, \boldsymbol{\Phi}, \mathbf{w}_{x,\tau}, \mathbf{w}_{y,\tau}\right), \quad (10)$$

where each conditional GP at the right side of the equation has a likelihood,

$$p\left(\mathbf{y}_k|\mathbf{Y}_{1:k-1}, \boldsymbol{\Phi}, \mathbf{w}_{x,\tau}, \mathbf{w}_{y,\tau}\right) = \mathcal{N}\left(\mathbf{y}_k|\boldsymbol{\Phi}^{\top} \mathbf{w}_{x,\tau} + \mathbf{Y}_{1:k-1}^{\top} \mathbf{w}_{y,\tau}, \sigma_k^2 I\right). \quad (11)$$

The prior distribution of each weight vector $\mathbf{w}_{x,\tau}$ is modeled as

$$p(\mathbf{w}_{x,\tau}) = \mathcal{N}\left(\mathbf{w}_{x,\tau}|\mathbf{0}, b_{\tau} \boldsymbol{\Sigma}_p\right), \quad (12)$$

and where for each task, a conditional one output likelihood GP is modeled with mean $\mathbf{w}_{y,\tau}^{\top} \mathbf{y}_{k,\tau-1}$ and covariance $b_{\tau} \mathbf{K}$ (see Figure 8d), where $[\mathbf{K}]_{k,l} = k(\hat{\mathbf{x}}_k, \hat{\mathbf{x}}_l)$.

To solve for primal weight vectors $\mathbf{w}_{y,\tau}$, we define a prior of these parameters with zero mean and identity covariance matrix[45] and infer a posterior all parameters following the formulation of the standard

GPR[46]

$$p\left([\mathbf{w}_{x,\tau}, \mathbf{w}_{y,\tau}]|\boldsymbol{\Phi}, \mathbf{Y}_{1:\tau}\right) = \mathcal{N}\left(\begin{bmatrix} \mathbf{w}_{x,\tau} \\ \mathbf{w}_{y,\tau} \end{bmatrix} \middle| \begin{bmatrix} \bar{\mathbf{w}}_{x,\tau} \\ \bar{\mathbf{w}}_{y,\tau} \end{bmatrix}, \mathbf{A}_{\tau}^{-1}\right). \quad (13)$$

The posterior is proportional to the product of the prior times the model likelihood. Solving for matrix $\mathbf{A}$ and vector $\bar{\mathbf{w}}_{y,\tau}$ gives the solution, which has to be obtained through a dual formulation, provided that the observations are transformed into space $\mathcal{H}$.

After this, parameters $b_\tau, \sigma_k^2$ and the kernel parameters are solved by maximizing the joint log-likelihood over all tasks. Once this optimization is done, the solution for $\mathbf{w}_{x,\tau}$ is given in dual form as $\bar{\mathbf{w}}_{x,\tau} = \boldsymbol{\Phi}\boldsymbol{\alpha}_\tau$, where

$$\boldsymbol{\alpha}_\tau = \mathbf{K}_{x,\tau}^{-1}\left(\mathbf{y}_\tau - \bar{\mathbf{w}}_{y,\tau}^\top \mathbf{Y}_{1:\tau-1}\right). \quad (14)$$

In the equation, $\mathbf{K}_{x,\tau} = (b_\tau \mathbf{K} + \sigma_\tau^2 \mathbf{I})$, and $\mathbf{K}$ is the kernel matrix containing the kernel dot products $k(\mathbf{x}_k, \mathbf{x}_l)$ between samples.

## Data structure
The 48 h forecasting horizon provided by the HRRR is at 4 pm. The proposed day-ahead energy forecast is provided at 5 pm, assuming a 1 h lag. The data structure in the forecasting feature vectors is in this section.

**Feature vectors for sparse learning.** The feature vectors in the sparse models $\mathbf{x}_i$ are from the reanalysis dataset ($\mathcal{A}$); as found in the Processing and Filtering. We have a different multivariate feature vector for each energy feature: electricity demand ($\mathbf{x}_i^{\mathcal{L}}$), solar generation ($\mathbf{x}_i^{\mathcal{S}}$), and wind geHneration ($\mathbf{x}_i^{\mathcal{W}}$). The weather features have an image-like structure and are filtered, $\psi(\cdot)$, to reduce the spatial dimensions. The spatial mask $\psi^{\mathcal{L}}(\cdot)$ applied to the electricity demand is based on the population density. The spatial mask $\psi^{\mathcal{S}}(\cdot)$ applied to the solar features is based on the installed solar generation capacity, and the spatial mask $\psi^{\mathcal{W}}(\cdot)$ applied to the wind features represents the installed wind generation capacity. The spatial mask $\psi^{\mathcal{E}}(\cdot)$ contains the intersection of $\psi^{\mathcal{L}} \cap \psi^{\mathcal{S}} \cap \psi^{\mathcal{W}}$ for the nodel-level model.

The weather features in the vector for electricity demand $\mathbf{x}_i^{\mathcal{L}}$ is formed by $\mathbf{r}_k' = \psi^{\mathcal{L}}(\mathbf{r}_i)$ (DSWRF), $\mathbf{d}_i' = \psi^{\mathcal{L}}(\mathbf{d}_i)$ (dew point), $\mathbf{h}_i' = \psi^{\mathcal{L}}(\mathbf{h}_i)$ (relative humidity), $\boldsymbol{\tau}_i' = \psi^{\mathcal{L}}(\mathbf{t}_i)$ (temperature), and $\mathbf{p}_i' = \psi^{\mathcal{L}}(\mathbf{p}_i)$ (discomfort index). The dimensions of the resulting features are $\mathbf{r}_i', \mathbf{d}_i', \mathbf{h}_i', \boldsymbol{\tau}_i', \mathbf{p}_i' \in \mathbb{R}^{D_1^{\mathcal{L}}}$. The feature vector for electricity demand is,

$$\mathbf{x}_i^{\mathcal{L}} = \left[r_{i,1}' \cdots r_{i,D_1^{\mathcal{L}}}' d_{i,1}' \cdots d_{i,D_1^{\mathcal{L}}}' h_{i,1}' \cdots h_{i,D_1^{\mathcal{L}}}' \tau_{i,1}' \cdots \tau_{i,D_1^{\mathcal{L}}}' p_{i,1}' \cdots p_{i,D_1^{\mathcal{L}}}'\right], \forall i = 1, \ldots, K \cdot T. \quad (15)$$

Similarly, the features in the vector $\mathbf{x}_i^{\mathcal{S}}$ for solar generation are $\mathbf{i}_i' = \psi^{\mathcal{S}}(\mathbf{i}_i)$ (DLWRF), $\mathbf{r}_i' = \psi^{\mathcal{S}}(\mathbf{r}_i)$ (DSWRF), and $\mathbf{g}_i' = \psi^{\mathcal{S}}(\mathbf{g}_i)$ (GHI). The dimensions of the feature vector are $\mathbf{i}_i', \mathbf{r}_i', \mathbf{g}_i' \in \mathbb{R}^{D_1^{\mathcal{S}}}$. The resulting feature vector for solar generation is,

$$\mathbf{x}_i^{\mathcal{S}} = \left[i_{i,1}' \cdots i_{i,D_1^{\mathcal{S}}}' r_{i,1}' \cdots r_{i,D_1^{\mathcal{S}}}' g_{i,1}' \cdots g_{i,D_1^{\mathcal{S}}}'\right], \forall i = 1, \ldots, K \cdot T. \quad (16)$$

The feature vector $\mathbf{x}_i^{\mathcal{W}}$ for wind generation contains wind speed features: $\boldsymbol{\omega}_i'^{60} = \psi^{\mathcal{W}}(\boldsymbol{\omega}_i'^{60})$ (wind speed at 60 m), $\boldsymbol{\omega}_i'^{80} = \psi^{\mathcal{W}}(\boldsymbol{\omega}_i'^{80})$ (wind speed at 80 m), $\boldsymbol{\omega}_i'^{100} = \psi^{\mathcal{W}}(\boldsymbol{\omega}_i'^{100})$ (wind speed at 100 m), and $\boldsymbol{\omega}_i'^{120} = \psi^{\mathcal{W}}(\boldsymbol{\omega}_i'^{120})$ (wind speed at 120 m), so that $\boldsymbol{\omega}_i'^{60}, \boldsymbol{\omega}_i'^{80}, \boldsymbol{\omega}_i'^{100}, \boldsymbol{\omega}_i'^{120} \in \mathbb{R}^{D_1^{\mathcal{W}}}$. The features in the vector for wind generation are,

$$\mathbf{x}_i^{\mathcal{W}} = \left[\omega_{i,1}'^{60} \cdots \omega_{i,D_1^{\mathcal{W}}}'^{60} \omega_{i,1}'^{80} \cdots \omega_{i,D_1^{\mathcal{W}}}'^{80} \omega_{i,1}'^{100} \cdots w_{i,D_1^{\mathcal{W}}}'^{100} \omega_{i,1}'^{120} \cdots \omega_{i,D_1^{\mathcal{W}}}'^{120}\right], \forall i = 1, \ldots, K \cdot T. \quad (17)$$

**Pattern vectors for Bayesian learning.** The feature vectors in for Bayesian learning step are the same as in the sparse learning step but

are from the forecast dataset ($\mathcal{F}$) instead of the reanalysis dataset ($\mathcal{A}$); as found in the Processing and Filtering. Therefore, we have a different feature vector for each energy feature: $\hat{\mathbf{x}}_{k,t}^{\mathcal{L}}$ (electricity demand), $\hat{\mathbf{x}}_{k,t}^{\mathcal{S}}$ (solar generation) and $\hat{\mathbf{x}}_{k,t}^{\mathcal{W}}$ (wind generation). The feature vectors contain information for each sample $k$ (days) and forecasting horizon $t$ (day hour), so the data structure is different. The forecasted weather features have an additional filtering step, $\delta(\cdot)$, that utilizes the coefficients from the sparse learning methods $\hat{\mathbf{x}}_{k,t}^{\mathcal{X}} = \{\hat{x}_{k,t,d}^{\mathcal{X}} | \sum_{z=1}^Z |\hat{w}_{d,z}^{\mathcal{X}}| > 0, \forall d = 1, \ldots, D_1^{\mathcal{X}}\}$ for each independent energy feature ($\mathcal{X} \in \{\mathcal{L}, \mathcal{S}, \mathcal{W}\}$).

The features in the forecasting vector for electricity demand ($\hat{\mathbf{x}}_{k,t}^{\mathcal{L}}$) are: $\hat{\mathbf{r}}_{k,t}'' = \delta(\hat{\mathbf{r}}_{k,t}')$, $\hat{\mathbf{d}}_{k,t}'' = \delta(\hat{\mathbf{d}}_{k,t}')$, $\hat{\mathbf{h}}_{k,t}'' = \delta(\hat{\mathbf{h}}_{k,t}')$, $\hat{\boldsymbol{\tau}}_{k,t}'' = \delta(\hat{\boldsymbol{\tau}}_{k,t}')$, and $\hat{\mathbf{p}}_{k,t}'' = \delta(\hat{\mathbf{p}}_{k,t}')$, $\hat{\mathbf{r}}_{k,t}'', \hat{\mathbf{d}}_{k,t}'', \hat{\mathbf{h}}_{k,t}'', \hat{\boldsymbol{\tau}}_{k,t}'', \hat{\mathbf{p}}_{k,t}'' \in \mathbb{R}^{D_2^{\mathcal{L}}}$. The electricity demand vector includes $\mathbf{y}_{k-\ell_1,t}^{\mathcal{L}}$ demand observations of the $\ell_1$ past days at hour $t$, and $\ell_2$ from the same operational day $\mathbf{y}_{k,t-\ell_2}^{\mathcal{L}}$. The vector also has temporal features (Supplementary Note 6): $z_{k,1}$ (year), $z_{k,2}$ (year day), and $z_{k,3}$ (week day); and auxiliary variables $z_{k,4}$ (weekend), $z_{k,5}$ (Holiday), $z_{k,6}$ (daylight saving time), and $t$ (day hour),

$$\hat{\mathbf{x}}_{k,t}^{\mathcal{L}} = \left[\hat{r}_{k,t,1}'' \cdots \hat{r}_{k,t,D_2^{\mathcal{L}}}'' \hat{d}_{k,t,1}'' \cdots \hat{d}_{k,t,D_2^{\mathcal{L}}}'' \hat{h}_{k,t,1}'' \cdots \hat{h}_{k,t,D_2^{\mathcal{L}}}'' \hat{\tau}_{k,t,1}'' \cdots \hat{\tau}_{k,t,D_2^{\mathcal{L}}}'' \right.$$
$$\left. \tilde{p}_{k,t,1}'' \cdots \hat{p}_{k,t,D_2^{\mathcal{L}}}'' y_{k-\ell_1,t}^{\mathcal{L}} y_{k-1,t}^{\mathcal{L}} y_{k,t-\ell_2}^{\mathcal{L}} \cdots y_{k,t-1}^{\mathcal{L}} z_{k,1} \cdots z_{k,6} t\right]. \quad (18)$$

The features in vector ($\hat{\mathbf{x}}_{k,t}^{\mathcal{S}}$) to forecast solar generation are $\hat{\mathbf{i}}_{k,t}'' = \delta(\hat{\mathbf{i}}_{k,t}')$, $\hat{\mathbf{r}}_{k,t}'' = \delta(\hat{\mathbf{r}}_{k,t}')$, and $\hat{\mathbf{g}}_{k,t}'' = \delta(\hat{\mathbf{g}}_{k,t}')$, $\hat{\mathbf{r}}_{k,t}'', \hat{\mathbf{i}}_{k,t}'', \hat{\mathbf{g}}_{k,t}'' \in \mathbb{R}^{D_2^{\mathcal{S}}}$. In addition, the feature vector includes time series from $\mathbf{y}_{k-\ell_1,t}^{\mathcal{S}}$ actual solar generation of $\ell_1 = 6$ past operational days, and past $\ell_2 = 16$ day hours $\mathbf{y}_{k,t-\ell_2}^{\mathcal{S}}$ and temporal features $z_{k,1}$, $z_{k,2}$, and $z_{k,6}$,

$$\hat{\mathbf{x}}_{k,t}^{\mathcal{S}} = \left[\hat{i}_{k,t,1}'' \cdots \hat{i}_{k,t,D_2^{\mathcal{S}}}'' \hat{r}_{k,t,1}'' \cdots \hat{r}_{k,t,D_2^{\mathcal{S}}}'' \hat{g}_{k,t,1}'' \cdots \hat{g}_{k,t,D_2^{\mathcal{S}}}'' y_{k-\ell_1,t}^{\mathcal{S}} \cdots y_{k-1,t}^{\mathcal{S}} \right.$$
$$\left. y_{k,t-\ell_2}^{\mathcal{S}} \cdots y_{k,t-1}^{\mathcal{S}} z_{k,1} z_{k,2} z_{k,6}\right]. \quad (19)$$

The features in the vector to forecast wind generation ($\hat{\mathbf{x}}_{k,t}^{\mathcal{W}}$) are: $\hat{\mathbf{w}}_{k,t}''^{60} = \delta(\hat{\mathbf{w}}_{k,t}'^{60})$, $\hat{\mathbf{w}}_{k,t}''^{80} = \delta(\hat{\mathbf{w}}_{k,t}'^{80})$, $\hat{\mathbf{w}}_{k,t}''^{100} = \delta(\hat{\mathbf{w}}_{k,t}'^{100})$, and $\hat{\mathbf{w}}_{k,t}''^{120} = \delta(\hat{\mathbf{w}}_{k,t}'^{120})$, $\hat{\mathbf{w}}_{k,t}''^{60}, \hat{\mathbf{w}}_{k,t}''^{80}, \hat{\mathbf{w}}_{k,t}''^{100}, \hat{\mathbf{w}}_{k,t}''^{120} \in \mathbb{R}^{D_2^{\mathcal{W}}}$. Similarly, the wind generation vector includes $\mathbf{y}_{k-\ell_1,t}^{\mathcal{W}}$ observed wind energy generation from the $\ell_1$ past days at the $t$ time of the day, and $\ell_2$ from the same day $\mathbf{y}_{k,t-\ell_2}^{\mathcal{W}}$. The vector has the same temporal features and auxiliary variables as the solar generation pattern $\hat{\mathbf{x}}_{k,t}^{\mathcal{S}}$. The forecast vector for wind generation is

$$\hat{\mathbf{x}}_{k,t}^{\mathcal{W}} = \left[\hat{w}_{k,t,1}''^{60} \cdots \hat{w}_{k,t,D_2^{\mathcal{W}}}''^{60} \hat{w}_{k,t,1}''^{80} \cdots \hat{w}_{k,t,D_2^{\mathcal{W}}}''^{80} \hat{w}_{k,t,1}''^{100} \cdots \hat{w}_{k,t,D_2^{\mathcal{W}}}''^{100} \right.$$
$$\left. \hat{w}_{k,t,1}''^{120} \cdots \hat{w}_{k,t,D_2^{\mathcal{W}}}''^{120} y_{k-\ell_1,t}^{\mathcal{W}} \cdots y_{k-1,t}^{\mathcal{W}} y_{k,t-\ell_2}^{\mathcal{W}} \cdots y_{k,t-1}^{\mathcal{W}} z_{k,1} z_{k,2} z_{k,6}\right]. \quad (20)$$

The energy vector includes non-repeated features from the electricity demand $\hat{\mathbf{x}}_{k,t,z}^{\mathcal{L}}$, solar generation $\hat{\mathbf{x}}_{k,t,z}^{\mathcal{S}}$, and wind generation $\hat{\mathbf{x}}_{k,t,z}^{\mathcal{W}}$ vectors. The filter $\delta(\cdot)$ applied to the joint energy features $\hat{\mathbf{x}}_{k,t,z}^{\mathcal{X}} = \{\hat{x}_{k,t,d,z}^{\mathcal{X}} | |\hat{w}_{d,z}^{\mathcal{X}}| > 0, \forall d = 1, \ldots, D_1^{\mathcal{X}}\}$ selects weather features independently for electricity demand ($\mathcal{L}$), solar generation ($\mathcal{S}$) and wind generation ($\mathcal{W}$), and does not sum the sparse learning coefficients $\hat{w}_{d,z}^{\mathcal{X}}$ across nodes $z$ (NP15, SP15, ZP26). A different energy feature vector $\hat{\mathbf{x}}_{k,t,z}^{\mathcal{E}}$ exists for each node. The nodal-level energy feature vectors are,

$$\hat{\mathbf{x}}_{k,t,z}^{\mathcal{E}} = \left[\hat{\mathbf{x}}_{k,t,z}^{\mathcal{L}} \hat{\mathbf{x}}_{k,t,z}^{\mathcal{S}} \hat{\mathbf{x}}_{k,t,z}^{\mathcal{W}}\right]. \quad (21)$$

The parameters $\ell_1$ and $\ell_2$ are set to $\ell_1 = \ell_2 = 6$ for all the proposed feature vectors ($\hat{\mathbf{x}}_{k,t}^{\mathcal{L}}, \hat{\mathbf{x}}_{k,t}^{\mathcal{S}}, \hat{\mathbf{x}}_{k,t}^{\mathcal{W}}$, and $\hat{\mathbf{x}}_{k,t,z}^{\mathcal{E}}$).

**Model chain.** We propose a model based on a conditional one-output likelihood multi-task GPR formulation[45],

$$y_{k,t} = \varphi^\top(\widehat{\mathbf{x}}^\star_{k,t})\mathbf{w}_{\widehat{\mathbf{x}},t} + \mathbf{y}^\top_{k,1:t-1}\mathbf{w}_{\mathbf{y},t} + \varepsilon_{k,t}, \tag{22}$$

where each solar generation forecasting horizon $y_{k,t}$ (task) is conditional to previous forecasting horizon $\mathbf{y}_{k,1:t-1}$ (tasks).

## Scenario smoothing

The predictive probability density for a new sample ($\star$), contains predictions from previous models in the chain $p(\widehat{y}^\star_{t,z}|\mathcal{D}_z, \widehat{\mathbf{x}}^\star_{t,z}, \widehat{y}^\star_{1,z}, \dots \widehat{y}^\star_{t-1,z})$, but not previous predictions $\widehat{\mathbf{x}}^\star_{1,z}, \dots, \widehat{\mathbf{x}}^\star_{t-1,z}$ (as found in the Model Chain). Due to this partially independent assumption on the weather forecasts from previous time instants, the shape of the scenario is not completely realistic despite representing the full probabilistic density (i.e., VS is low). To overcome this limitation, we apply to predictive scenarios a 1-dimensional Gaussian convolutional kernel to smooth them and reduce oscillations introduced by discretizations,

$$\widetilde{y}^\star_{t,z,s} = \sum_{t'=1}^T \widehat{y}^\star_{t',z,s}\left(\frac{1}{\sqrt{2\pi}\sigma_z}\exp\left\{-\frac{(t'-t)^2}{2\sigma^2_z}\right\}\right), \forall t = 1, \dots, T, \tag{23}$$

where $t' = 1, \dots, T$ is the scalar corresponding to hour. The convolutional kernel is applied to the scenarios $s = 1, \dots, S$ generated from the predictive posteriors $\widehat{y}^\star_{t,z,s} \sim p(\widehat{y}^\star_{t,z}|\mathcal{D}_z, \widehat{\mathbf{x}}^\star_{t,z}, \widehat{y}^\star_{1,z}, \dots \widehat{y}^\star_{t-1,z})$. The parameter $\sigma_z$ is different for each energy feature ($\mathcal{L}, \mathcal{S}$ and $\mathcal{W}$) and zone $z$, and requires cross-validation.

## Predictive density calibration

The Bayesian methods predict the mean and covariance from the predictive density function. However, when validating confidence intervals, we found that the MTGPR does not produce a predictive covariance matrix under all circumstances. This is a common problem in other formulations of an MTGPR[82,83]. In particular, we observed a constant bias on the predictive covariance function on the MTGPR model when forecasting different energy features, electricity demand ($\widehat{\mathbf{y}}^\mathcal{L}$), solar generation ($\widehat{\mathbf{y}}^\mathcal{S}$) and wind generation ($\widehat{\mathbf{y}}^\mathcal{L}$) from the same node ($z = 1$ for instance). The MTGPR model does not have this limitation when predicting the same energy feature at different nodes, for solar generation at NP15 ($\widehat{\mathbf{y}}^\mathcal{S}_{z=1}$), SP15 ($\widehat{\mathbf{y}}^\mathcal{S}_{z=2}$) and ZP26 ($\widehat{\mathbf{y}}^\mathcal{S}_{z=3}$).

To unbias the predictive covariance matrix, we first calculate the true predictive covariance matrix $\boldsymbol{\Gamma}_{k,t,z}$ for a sample $\mathbf{y}_{k,t,z}$ given that the predictive mean is $\widehat{\boldsymbol{\mu}}_{k,t,z}$,

$$\boldsymbol{\Gamma}_{k,t,z} = (\mathbf{y}_{k,t,z} - \widehat{\boldsymbol{\mu}}_{k,t,z})^\top(\mathbf{y}_{k,t,z} - \widehat{\boldsymbol{\mu}}_{k,t,z}). \tag{24}$$

Then, we propose the following model to unbias the predictive covariance matrix $\widehat{\boldsymbol{\Sigma}}_{k,t,z}$,

$$\begin{aligned}\boldsymbol{\gamma}_{k,t,z} &= \mathbf{w}^\top_{t,z}\widehat{\mathbf{s}}_{k,t,z} + \mathbf{e}_{k,t,z} \\ \widehat{\mathbf{w}}_{t,z} &= (\widehat{\mathbf{S}}^\top_{t,z}\widehat{\mathbf{S}}_{t,z})^{-1}\widehat{\mathbf{S}}^\top_{t,z}\boldsymbol{\Gamma}_{t,z} \\ \widehat{\boldsymbol{\gamma}}^\star_{t,z} &= \widehat{\mathbf{w}}^\top_{t,z}\widehat{\mathbf{s}}^\star_{t,z}\end{aligned} \tag{25}$$

where $\widehat{\mathbf{s}}_{k,t,z} = [\text{vec}(\widehat{\boldsymbol{\Sigma}}_{k,t,z})\,1]$, $\boldsymbol{\gamma}_{k,t,z} = \text{vec}(\boldsymbol{\Gamma}_{k,t,z})$, and $\widehat{\boldsymbol{\gamma}}^\star_{t,z}$ is the unbiased covariance matrix in vector form for a new sample ($\star$) at the hour $t$ in the node $z$. The predictive probability distribution with the unbiased covariance matrix $\widehat{\boldsymbol{\Gamma}}^\star_{t,z} = \text{vec}^{-1}(\widehat{\boldsymbol{\gamma}}^\star_{t,z})$ is now $\mathcal{N}(\widehat{\boldsymbol{\mu}}^\star_{t,z}, \widehat{\boldsymbol{\Gamma}}^\star_{t,z}) \sim p(\widehat{\mathbf{y}}^\star_{t,z}|\mathcal{D}_z, \widehat{\mathbf{x}}^\star_{t,z}, \widehat{y}^\star_{1,z}, \dots \widehat{y}^\star_{t-1,z})$. $\text{vec}(\cdot)$ represents the vectorization of a matrix, and $\text{vec}^{-1}(\cdot)$ restores the vector to its original matrix shape.

## Energy imbalance

The operational reserves are a statistic of the historical magnitude of the errors in the day-ahead forecast. In particular, the fraction $\Delta^{\text{up}}$ of up reserves is 6% of the day-ahead forecasted electricity demand at each time point in CAISO[55], 50% must be spinning reserves. The system must have enough reserves to supply non-delivered contracted imports and guarantee the supply of contracted exports[56]. In addition, the current trend follows an increase in regulation down reserves equivalent to the regulation up plus spinning reserves combined, and an increase in non-spinning, so that regulation up, regulation down plus spinning reserves are approximately equal to the non-spinning[12]. Therefore, we assume a total estimation for the downward reserves is $\Delta^{\text{down}} \approx 0.12$ (12%), and that the total upward and downward reserves are the same $\Delta^{\text{up}} = \Delta^{\text{down}} = \Delta$ for simplicity.

The reserve levels are $r^{\text{CAISO}\star}_t = (\Delta^{\text{up}} + \Delta^{\text{down}}) \cdot \sum_{z=1}^Z \widehat{y}^{\mathcal{L}\star}_{\text{CAISO},t,z} = 2\Delta \cdot \sum_{z=1}^Z \widehat{y}^{\mathcal{L}\star}_{\text{CAISO},t,z}$. We want to find the z-interval $\varrho$ that defines reserve levels $r^{\text{ML}\star}_{t,z} = 2\varrho \cdot \sum_{z=1}^Z (\widehat{\sigma}^{\mathcal{L}\star^2}_{t,z} + \widehat{\sigma}^{\mathcal{S}\star^2}_{t,z} + \widehat{\sigma}^{\mathcal{W}\star^2}_{t,z})$ with the same capacity that $r^{\text{CAISO}\star}_t$, but allocated according to the variance on the predictive posterior aggregated across nodes (assuming independence between $\mathcal{L}, \mathcal{S}$ and $\mathcal{W}$). We propose to derive the z-interval $\varrho$ from this equivalence,

$$\sum_{t=1}^T r^{\text{CAISO}\star}_t = \sum_{t=1}^T r^{\text{ML}\star}_t$$
$$\varrho = \frac{2\Delta \cdot \sum_{t=1}^T \sum_{z=1}^Z \widehat{y}^{\mathcal{L}\star}_{\text{CAISO},t,z}}{2\sum_{t=1}^T \sum_{z=1}^Z (\widehat{\sigma}^{\mathcal{L}\star^2}_{t,z} + \widehat{\sigma}^{\mathcal{S}\star^2}_{t,z} + \widehat{\sigma}^{\mathcal{W}\star^2}_{t,z})}. \tag{26}$$

confidence bounds of the net demand $\widehat{y}^\mathcal{N}\star$ defined as $\left[\sum_{z=1}^Z \widehat{y}^{\mathcal{N}\star}_{\text{CAISO},t,z} - 0.5 \cdot r^{\text{CAISO}\star}_t, \sum_{z=1}^Z \widehat{y}^{\mathcal{N}\star}_{\text{CAISO},t,z} + 0.5 \cdot r^{\text{CAISO}\star}_t\right]$ for CAISO's, and equivalently $\left[\sum_{z=1}^Z \widehat{\mu}^{\mathcal{N}\star}_{t,z} - 0.5 \cdot r^{\text{ML}\star}_t, \sum_{z=1}^Z \widehat{\mu}^{\mathcal{N}\star}_{t,z} + 0.5 \cdot r^{\text{ML}\star}_t\right]$ for the alternative approach based on a probabilistic forecast. The net demand is $\widehat{y}^{\mathcal{N}\star}_{\text{CAISO},t,z} = \widehat{y}^{\mathcal{L}\star}_{\text{CAISO},t,z} - \widehat{y}^{\mathcal{S}\star}_{\text{CAISO},t,z} - \widehat{y}^{\mathcal{W}\star}_{\text{CAISO},t,z}$ for CAISO forecast and $\widehat{\mu}^\mathcal{L} = \widehat{\mu}^{\mathcal{L}\star}_{t,z} - \widehat{\mu}^{\mathcal{S}\star}_{t,z} - \widehat{\mu}^{\mathcal{W}\star}_{t,z}$ for the probabilistic forecast.

Operators participate in the real-time imbalance markets to balance energy supply with demand. Assuming that a grid operator commits enough capacity to supply the lower confidence interval $l^\star_t$ and can regulate the dispatch up to $u^\star_t$, the operator will interact with the market to buy $\iota^\star_t > 0$ or curtail energy $\iota^\star_t < 0$ at each time $t$ in response to the net demand $y^\mathcal{N}\star_t$ so that,

$$\iota^\star_t = \begin{cases} y^\mathcal{N}\star_t - u^\star_t & y^\mathcal{N}\star_t > u^\star_t \\ 0 & l^\star_t \leq y^\mathcal{N}\star_t \leq u^\star_t \\ l^\star_t - y^\mathcal{N}\star_t & y^\mathcal{N}\star_t < l^\star_t. \end{cases} \tag{27}$$

Therefore, if the net demand $y^\mathcal{N}\star_t$ is $l^\star_t \leq y^\mathcal{N}\star_t \leq u^\star_t$, then the system is balanced. If $y^\mathcal{N}\star_t > u^\star_t$, operators need to import energy from the imbalance market, and if $y^\mathcal{N}\star_t < l^\star_t$, an operator needs to curtail renewable energy (i.e., solar or wind generation).

## Experimental setup

**Data preprocessing.** While the HRRR data is available from Sep. 30, 2014, the OASIS data structure is consistent only from Jun. 30, 2019. Thus, our dataset includes days from Jun. 30, 2019, to Feb. 11, 2023. 17% of the samples (184 days) have at least a missing entry (an hour) either in the HRRR or OASIS database. We excluded these samples from the datasets ($\mathcal{A}$ and $\mathcal{F}$). After reducing the resolution (as found in the

Processing and Filtering), each weather feature has $108 \times 88$ spatial dimensions (9,152).

The weather vectors from the HRRR include 5 weather features for electricity demand, 3 for solar generation, and 4 for wind generation. Therefore, the feature vector for electricity demand has $5 \times 108 \times 88$ dimensions (47,520), the vector for solar generation has $3 \times 108 \times 88$ dimensions (27,456), the one for wind generation has $4 \times 108 \times 88$ dimensions (38,016), and node-level has $11 \times 108 \times 88$ dimensions (104,544) before applying the spatial filtering. The covariates and dependent variable are standardized (see Section Data Structure), so that    $\bar{y}_{i,t} = (y_{i,t} - \mathbb{E}[\mathbf{y}_t])/\mathbb{V}^{1/2}[\mathbf{y}_t]$,    and    $\bar{x}_{i,j} = (x_{i,j} - \mathbb{E}[\mathbf{x}_j])/\mathbb{V}^{1/2}[\mathbf{x}_j]$, which implies $\sum_{i=1}^{N} \bar{y}_{i,t} = \sum_{i=1}^{N} \bar{x}_{i,j} = 0$, and $\sum_{i=1}^{N} \bar{y}_{i,t}^2 = \sum_{i=1}^{N} \bar{x}_{i,j}^2 = N$.

**Validation, training, and testing.** The datasets $\mathcal{A}$ and $\mathcal{F}$ were divided into training (75%, or 729 days) and testing (25%, or 243 days) sets, maintaining the original time structure of the data (i.e., without random sampling). The hyperparameter's cross-validation was performed on the training set, implementing a $k$-fold cross-validation method. The number of folds was set to $k = 5$ to limit the computational time. After the cross-validation, the model is trained using the entire training set with a fixed set of optimal hyperparameters. The optimal hyperparameter selection criterion (i.e., model selection) is the lowest ES in cross-validation. The performances are evaluated on the testing set. This investigation implements four multivariate proper scoring rules to evaluate different characteristics of a probabilistic forecast: Energy Score (ES), Variogram Score with $p = 0.5$ ($VS^{0.5}$), and Interval Score (IS)[47].

The proper scoring rules are evaluated in the solar hours of the day to avoid numerical problems with non-zero entries (8 am to 4 pm). In the case of the MTGPR with all energy features (electricity demand, solar generation, and wind generation), the proper scoring rules are evaluated at three different time intervals to avoid this problem: midnight-sunrise (0 am to 7 am), daylight (8 am to 4 pm), and sunset-midnight (5 pm to 11 pm). We evaluate the electricity demand and wind generation forecast in the midnight-sunrise and sunset-midnight periods, and the electricity demand, solar generation, and wind generation forecast during daylight. The scores of each period are added together to obtain the final score used for the evaluation.

**Hyperparameters.** The sparse learning methods have hyperparameters that require cross-validation. Lasso hyperparameter $\lambda$, swept between $10^{-4}$ and 10, in equation (2) adjusts a trade-off in the $L_1$-norm. $\beta$, swept between 10 and 640, in equation (3), defines the number of coefficients. The hyperparameters in Elastic Net that define the trade-off between the $L_1$-norm and $L_2$-norm in equation (4) are cross-validated as $\Omega_1 = \kappa\rho$ and $\Omega_2 = (1 - \kappa)\rho$, $\kappa$ is swept between $10^{-4}$ and 1, and $\rho$ between 0.01 and 0.75. The hyperparameters in equation (5) set the trade-off between the $L_1$-norm and the group $L_2$-norm for Group Lasso, which are defined as $\xi_2 = (1 - \eta)\chi$ and $\xi_1 = \eta\chi$, $\chi$ is swept between 1 and 100, $\eta$ between 0.25 and 1.

The Bayesian learning methods also have hyperparameters that require cross-validation. We cross-validate the ARD hyperparameter $\gamma_0$ between $10^{-4}$ and 100; as found in Relevance Vector Machine (RVM), and the kernel functions in the GPR and MTGPR (Supplementary Note 2): Linear ($L$), Rational Quadratic ($RQ$), and Matérn ($M_{0.5}$, $M_{1.5}$, and $M_{2.5}$). The hyperparameters in BLR are initialized to $\alpha_n = \beta_n = \alpha_p = \beta_p = 10^{-6}$ (non-informative) and they do not require cross-validation; as found in Bayesian Linear Regression (BLR). The BLR formulation is not equivalent to a GPR with a linear kernel ($\mathcal{K}_L$). $\mathcal{K}_L$ has amplitude $\theta_1$ and bias $\theta_2$ hyperparameters (Supplementary Note 2).

The criterion for selecting the hyperparameters shown in the results of this article is the lowest ES in validation. When the criterion is

lower $VS^{0.5}$ or IS, the selected models may be different, see Supplementary Figs. 11-16.

**Predictive scenarios smoothing and density calibration.** The approaches proposed to smooth the generated scenarios (see Section Scenario Smoothing) have a parameter $\sigma_z$ in equation (23). This parameter requires cross-validation for each node $z$ and energy feature ($\mathcal{L}$, $\mathcal{S}$, and $\mathcal{W}$). The testing set in the cross-validation is used to fine-tune $\sigma_z$, sweeping values from 0 to 1.25. The shape of the scenarios is evaluated with the ES and the $VS^{0.5}$, and the smaller $\sigma_z$ parameter with lower ES or $VS^{0.5}$ is selected. The optimal $\sigma_z$ is the average across the optimal values obtained for each 5-fold interaction.

The calibration of the predictive covariance in an MTGPR with all energy features requires estimating the parameters $\hat{\mathbf{w}}_{t,z}$ for each hour $t$ at each node $z$; see equation (25). In each interaction of the cross-validation, the testing set is used to find the optimal parameters $\hat{\mathbf{w}}_{t,z}$. The optimal $\hat{\mathbf{w}}_{t,z}$ is the average across the optimal values obtained for each 5-fold interaction. This approach is based on the conformal learning literature[51,52].

The forecasting models are evaluated without smoothing or calibration to avoid overfitting issues. Only testing scores are calculated with smoothed scenarios and calibrated predictive covariance. A potentially better praxis is to split the validation set into training, calibration, and testing. However, we do not have enough observations to implement this approach.

**Computing resources.** The experiments were performed in POD, a cluster computer maintained by the Center for Scientific Computing (CSC) at the UC Santa Barbara. POD has 70 nodes with a Dual Intel Xeon Gold 6148 Processor at 2.40 GHz. Each node has 20 CPUs, 40 threads, and 187 GB of RAM.

## Data availability
The processed data necessary to replicate the experiments with the developed software is in a Zenodo data repository[85]. The raw data is not publicly available due to large size, but access can be obtained by upon request. The results data generated in this study is provided in the Supplementary Tables.

## Code availability
The software developed for the experiments is publicly available in a GitHub repository[86]. The software for the MTGPR is also in a publicly available GitHub repository (github.com/OGHinde/Cool_MTGP). The GPR software was developed using the GPyTorch library (gpytorch.ai). The software for sparse learning and other Bayesian learning methods (BLR and RVM) is based on the Scikit-Learn library (scikit-learn.org). The additional utilities for downloading, processing, and visualizing data are also publicly available in the GitHub repository[87].

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

## Acknowledgements

R.D. and G.T.S were supported by the Research Seed Grant Program of the Institute for Energy Efficiency (IEE) and the Climate Innovation Funds provided by the California NanoSystems Institute (CNSI) at the UC Santa Barbara (UCSB). M.M.R. has been partially supported by the King Felipe VI Endowed Chair of the University of New Mexico. G.T.S was supported with the Climate Innovation Postdoctoral Fellowship from CNSI at UCSB. G.T.S. thanks Prof. Ludkovski for providing valuable insights. Use was made of the computational facilities purchased with funds from the National Science Foundation (CNS-1725797) and administered by the Center for Scientific Computing (CSC). The CSC is supported by the CNSI and the Materials Research Science and Engineering Center (MRSEC; NSF DMR 2308708) at UCSB.

## Author contributions

R.D. and G.T.S. conceptualized the study, acquired funding, work on the visualization. R.D. and M.M.R. supervised the project. R.D., M.M.R., and G.T.S. developed the methodology. R.D. did the project administration. G.T.S. developed the software, curated the data, and drafted the manuscript. All authors worked in the formal analyses and investigation, reviewed, edited, and approved the final manuscript.

## Competing interests

The authors declare no competing interests.
