## [Transparent Peer Review file · Nature Communications]

Probabilistic Day-Ahead Forecasting of System-Level Renewable Energy and Electricity Demand

Corresponding Author: Dr Guillermo Terrén-Serrano

Version 0:

Reviewer comments:

Reviewer #1

(Remarks to the Author)

- What are the noteworthy results?

Numerical analysis and comparison of joint probabilistic forecasting with several benchmarks.

- Will the work be of significance to the field and related fields? How does it compare to the established literature?

The topic is timely and relevant, and the manuscript is well written. Although the work is original with respect to the existing literature regarding joint probabilistic forecast (for example in comparison with existing literature on solar probabilistic forecasts with Bayesian methods), the work is not that significant to the field of energy system as the case study is limited to the specific case of California ISO. Moreover, some important aspects on the performance and quality of forecasting methods are not sufficiently discussed. This includes for example the level of aggregation of demand and generation, demand configuration (e.g., residential vs industrial), the share of weather sensitive loads (e.g., cooling and heating systems) on total demand, and forecasting horizon.

- Does the work support the conclusions and claims, or is additional evidence needed?

The claims as declared in the title and abstract are vast and promising while the overall results are limited. Indeed, the title does not reflect the main contents of the paper.

- Are there any flaws in the data analysis, interpretation and conclusions? Do these prohibit publication or require revision?

A revision might be required. The robustness of the proposed method for joint probabilistic forecast against the error of weather forecast is not clearly analysed. Moreover, the application to the power system operation could be further analysed for example regarding the impact of spatial correlations on both demand and generation forecast error on grid congestions.

- Is the methodology sound? Does the work meet the expected standards in your field?

The methodology sounds.

- Is there enough detail provided in the methods for the work to be reproduced?

The methodology as described in the manuscript does not have enough details to ensure reproducibility, but the authors have provided the original codes. Adding comments and references in the code to the manuscript could better ensure reproducibility.

(Remarks on code availability)

Reviewer #2

(Remarks to the Author)

The authors propose a novel framework for probabilistic day-ahead forecasting of electricity demand and renewable energy generation (wind and solar) at the system level, using publicly available weather forecast data. The framework integrates sparse learning methods such as Lasso, OMP, and elastic net for feature selection with Bayesian learning methods such as

Gaussian process regression, relevance vector machines, and multi-Task GPR to quantify forecast uncertainty and jointly model demand and supply. The effectiveness of the proposed method is evaluated across three nodal regions in the CAISO system. Results show an improvement of up to 25.2% in net demand forecast accuracy compared to CAISO's operational forecast. The proposed method may have value for system level demand and renewable energy generation forecasting and uncertainty quantification. However, the reviewer thinks there are several areas where the clarity and reproducibility of the study can be improved. Therefore, I suggest a major revision.

1. The reviewer feels that the implementation details of the multi-task Gaussian process regression method is under described. For example, is the covariance kernel shared across tasks (I mean across locations or demand/VRE)? How the inter-task dependencies are learnt?
2. The ensemble method include Bayesian regression and sparse feature selection. It would be helpful to include ablation case study to showcase: (1) performance of Bayesian models with and without sparse feature selection; (2) impact of joint and independent modeling; (3) contribution of kernel selection.
3. As far as the reviewer known, in 2024, the Institute of Industrial & Systems Engineers (IISE), in partnership with Pacific Gas and Electric Company (PG&E), hosted an energy forecasting competition focused on nodal regions within CAISO. The reviewer has observed that many SOTA deep learning models—such as Temporal Fusion Transformer, Informer, and TimesNet—were widely adopted for day-ahead probabilistic forecasting in that context. It would be helpful for the authors to benchmark their approach against these models as well, particularly given their growing adoption by system operators and ISOs.
4. For reproducibility, the authors provide a GitHub repository; however, the reviewer was unable to locate the hyperparameter settings used in the experiments.
5. The reviewer believes the claim of “scalability to other ISOs” should be tempered unless the framework is validated on at least a couple of additional systems. CAISO is characterized by a uniquely high penetration of solar PV, pronounced diurnal demand patterns, and relatively mature data transparency and forecasting infrastructure. These features may not generalize to other ISO/RTOs, where renewable integration is often more wind-dominated, load patterns vary seasonally, and the availability and spatial granularity of weather and market data can differ significantly.
6. It would be helpful to clarify if the hyperparameters are optimized separately for each day or fixed across the test set?
7. It would be helpful to clarify how the group lasso is applied across grouped spatial features.

Some minor comments: In the supplementary document, it would be helpful to include a table summarizing the kernel parameters and their respective roles. Additionally, while the Logarithmic Score is introduced in the supplementary material, the reviewer did not observe its use in the main manuscript beyond its mention in the skill score calculation.

(Remarks on code availability)

For reproducibility, the authors provide a GitHub repository; however, the reviewer was unable to locate the hyperparameter settings used in the experiments. The readme file in https://github.com/gterren/caiso_power only include package information.

Reviewer #3

(Remarks to the Author)

The authors present a novel approach to joint probabilistic day-ahead forecasting of electricity demand, wind and solar. The manuscript is easy to follow and the contribution seems sound, but I'm no expert in these methods. I will therefore stick the following highlevel remarks:

- 1) The quality of the figures must be improved. Many labels are hard to read, and figures are often overloaded with information. Please relegate non-essential information to the appendix.
- 2) In Fig 1, you're plotting wind and solar variations over seasons, but what has this to do with probabilistic operating reserve sizing? It's day-ahead forecast errors that drive the need for operating reserves.
- 3) Figures 2 and onwards contain results from many methods. Would it be possible to select the best performing, and relegate the comparison to the supplementary material? This would help the readability.
- 4) The application to reserve sizing does not add value. The whole comparison hinges on some - in my opinion, questionable - assumptions on how to treat import/export. I would remove this section, and focus on the probabilistic forecasting. The contribution and relevance thereof is sufficient.

(Remarks on code availability)

Version 1:

Reviewer comments:

Reviewer #1

(Remarks to the Author)

In the revised manuscript, the authors have satisfactorily addressed this reviewer's comments and concerns.

(Remarks on code availability)

Yes, as mentioned by the authors, the README file is improved and relevant comments and references are added to the

code.

Reviewer #3

(Remarks to the Author)

Thank, you have addressed my comments.

(Remarks on code availability)

Manuscript Number: NY-610A-NPG&MTS.

Manuscript Title: Probabilistic Day-Ahead Forecasting of System-Level Renewable Energy and Electricity Demand.

NB: Commentaries of the reviewers are in standard style, modifications applied to the document are in bold style, and responses to the editors and reviewers are in italic style.

Point-by-point response to the reviewers' comments

Reviewer no. 1

1. What are the noteworthy results? Numerical analysis and comparison of joint probabilistic forecasting with several benchmarks.

We thank the reviewers for considering our results noteworthy.

2. Will the work be of significance to the field and related fields? How does it compare to the established literature? The topic is timely and relevant, and the manuscript is well written. Although the work is original with respect to the existing literature regarding joint probabilistic forecast (for example in comparison with existing literature on solar probabilistic forecasts with Bayesian methods), the work is not that significance to the field of energy system as the case study is limited to the specific case of California ISO. Moreover, some important aspects on the performance and quality of forecasting methods are not sufficiently discussed. This includes for example the level of aggregation of demand and generation, demand configuration (e.g., residential vs industrial), the share of weather sensitive loads (e.g., cooling and heating systems) on total demand, and forecasting horizon.

We agree with the reviewer that these limitations are not sufficiently discussed in the manuscript. Reviewer no. 2 raised a similar concern in question 5. We included this paragraph in the discussion section (see page 18 of the marked manuscript).

The results may vary when applied to other ISOs due to different regional demand patterns, generation mixes, and specific operational requirements (i.e., forecasting horizon and lag). However, the proposed joint day-ahead probabilistic forecasting method remains applicable and transferable with the appropriate datasets. Since our net demand forecast is based on a top-down approach that learns short-term patterns aggregated at the three main zones in CAISO, we implicitly model the bottom-up breakdown of disaggregated demand (e.g., residential, industrial, and weather-sensitive loads) and distributed generation. While this abstraction level may be insufficient for long-term disaggregated bottom-up demand forecasts, it is effective in our aggregated day-ahead net demand forecast application.

3. Does the work support the conclusions and claims, or is additional evidence needed? The claims as declared in the title and abstract are vast and promising while the overall results are limited. Indeed, the title does not reflect the main contents of the paper.

We highlighted several main features of the methodology in the abstract. The first one is related to the risk of forecasting error. We illustrate this feature in Section Probabilistic Day-Ahead Energy Forecast for Reserves Allocation. We explicitly address the statement in the manuscript to clarify that this is the feature discussed in the text that presents and discusses Fig. 6. Regarding the coordination of market operations, we do not prove in the paper that our methods improve this aspect of the electricity sector. Still, we believe that methodologies (as the ones presented in the paper) can be beneficial for this purpose. Therefore, we discuss the potential usefulness of probabilistic forecasts in improving coordination of market operations at the end of the section. To clarify the purpose of this discussion, we specifically mention wholesale electricity markets and forecasting errors.

4. Are there any flaws in the data analysis, interpretation and conclusions? Do these prohibit publication or require revision? A revision might be required. The robustness of the proposed method for joint probabilistic forecast against the error of weather forecast is not clearly analysed. Moreover, the application to the power system operation could be further analysed for example regarding the impact of spatial correlations on both demand and

generation forecast error on grid congestions.

We thank the reviewers for the comment. Reviewer no. 3 in question 4 had a similar concern. We added the following paragraphs to the Discussion/Conclusion section on page 18 of the marked manuscript. The intention is to inform the reader of the following step in this investigation, so they can understand the motivation behind proposing this new day-ahead forecast methodology. However, the methods and datasets required for the study are very different. They require justification and additional details that will make a single article very long and thus difficult to publish.

The proposed dynamic reserves allocation serves as an example of how to utilize a probabilistic forecast in power system operations. [...]

We intentionally integrated the ability to generate predictive scenarios associated with probabilities in a day-ahead forecast to assess the Conditional Value at Risk (CVaR) on dynamic reserves allocation and analyze the impact of spatial correlations on both demand and generation on grid congestion, considering congestion and committed units to account for energy imports and exports more accurately. However, this investigation will require methods and datasets beyond the scope of this study.

5. Is the methodology sound? Does the work meet the expected standards in your field? The methodology sounds.

We appreciate the reviewers' comment about the proposed methodology.

6. Is there enough detail provided in the methods for the work to be reproduced? The methodology as described in the manuscript does not have enough details to ensure reproducibility, but the authors have provided the original codes. Adding comments and references in the code to the manuscript could better ensure reproducibility.

We agree with the reviewers and thus we added further details to the README.md file in GitHub repository (github.com/gterren/caiso_power). In particular, we added comments in scripts and referenced sections in the manuscript. See:

- `preprocessing.py`
- `val_shallow_learning_batch.py`
- `test_shallow_learning_batch.py`
- `model_shallow_learning.py`
- `val_multisource_shallow_learning.py`
- `test_multisource_shallow_learning.py`
- `model_multisource_shallow_learning.py`

We also added the following information about the repositories in page 29 of the marked manuscript as Section Code Availability.

The software developed for the experiments is publicly available in a GitHub repository (github.com/gterren/caiso_power). The software for the MTGPR is also in a publicly available GitHub repository (github.com/OGHinde/Cool_MTGPR). The GPR software was developed using the GPyTorch library ([gpytorch.ai](https://github.com/pytorch/gpytorch)). The software for *sparse learning* and other *Bayesian learning* methods (BLR and RVM) is based on the Scikit-Learn library (scikit-learn.org). The additional utilities for downloading, processing, and visualizing data are also publicly available in the GitHub repository (github.com/gterren/caiso_power_viz).

We also shared the codes utilized in results visualization in a public GitHub repository (github.com/gterren/caiso_power_viz) and shared the results and datasets on a Zenodo data repository [TDM25]. We added this information to the Section Data Availability (see page 29 of the marked manuscript).

The datasets necessary to run the experiments with the codes in the GitHub repository github.com/gterren/caiso_power and visualize the results in the GitHub repository github.com/gterren/caiso_power_viz are available in a Zenodo data repository [TDM25].

Reviewer no. 2

The authors propose a novel framework for probabilistic day-ahead forecasting of electricity demand and renewable energy generation (wind and solar) at the system level, using publicly available weather forecast data. The framework integrates sparse learning methods such as Lasso, OMP, and elastic net for feature selection with Bayesian learning methods such as Gaussian process regression, relevance vector machines, and multi-Task GPR to quantify forecast uncertainty and jointly model demand and supply. The effectiveness of the proposed method is evaluated across three nodal regions in the CAISO system. Results show an improvement of up to 25.2% in net demand forecast accuracy compared to CAISO’s operational forecast. The proposed method may have value for system level demand and renewable energy generation forecasting and uncertainty quantification. However, the reviewer thinks there are several areas where the clarity and reproducibility of the study can be improved. Therefore, I suggest a major revision.

1. The reviewer feels that the implementation details of the multi-task Gaussian process regression method is under described. For example, is the covariance kernel shared across tasks (I mean across locations or demand/VRE)? How the inter-task dependencies are learnt?

We concur with the reviewer that the multi-task Gaussian process can be described in further detail to clarify all the omissions. We summarized the multi-task GPR proposed in [GMG22], adding the following text in pages 22-24 of the marked manuscript.

In a conditional one-output likelihood multi-task Gaussian process for regression (Cool-MTGPR) [GMG22], the task of estimating τ regressors $\mathbf{y}_k \in \mathbb{R}^T$ from predictor $\varphi(\hat{\mathbf{x}}_k)$ is done with the model

$$\mathbf{y}_{k,\tau} = \varphi(\hat{\mathbf{x}}_k)^\top \mathbf{w}_\tau + \varepsilon_{k,\tau} = \varphi(\hat{\mathbf{x}}_k)^\top \mathbf{w}_{\mathbf{x},\tau} + \mathbf{Y}_{k,1:\tau-1}^\top \mathbf{w}_{\mathbf{y},\tau} + \varepsilon_{k,\tau}. \quad (1)$$

In this formulation, each factorized task \mathbf{y}_k is modeled as dependent of the previous ones, and therefore the corresponding weights are split into $\mathbf{w}_{k,\tau} \in \mathcal{H}$ for the input sample $\hat{\mathbf{x}}_k$ and $\mathbf{w}_{\mathbf{y},\tau}$ for the previous tasks. Indeed, weight vectors \mathbf{w}_τ in a MTGPR can be recovered as $\mathbf{w}_\tau = \mathbf{w}_{\mathbf{x},\tau} + \mathbf{W}_{1:\tau-1} \mathbf{w}_{\mathbf{y},\tau}$. Here, model error ε is assumed to have the form of a Gaussian distribution $p(\varepsilon_k) \sim \mathcal{N}(\mathbf{0}, \Sigma_p)$.

By applying the chain rule of probability to the standard joint multitask likelihood, we can factorize it into a product of conditional probabilities, each one corresponding to each one of the conditional tasks in Eq. (1),

$$\mathbf{p}(\text{vect}(\mathbf{Y}) \mid \Phi, \mathbf{W}) = \prod_{k=1}^T \mathbf{p}(\mathbf{y}_k \mid \mathbf{Y}_{1:k-1}, \Phi, \mathbf{w}_{\mathbf{x},\tau}, \mathbf{w}_{\mathbf{y},\tau}), \quad (2)$$

where each conditional GP at the right side of the equation has a likelihood,

$$\mathbf{p}(\mathbf{y}_k \mid \mathbf{Y}_{1:k-1}, \Phi, \mathbf{w}_{\mathbf{x},\tau}, \mathbf{w}_{\mathbf{y},\tau}) = \mathcal{N}\left(\mathbf{y}_k \mid \Phi^\top \mathbf{w}_{\mathbf{x},\tau} + \mathbf{Y}_{1:k-1}^\top \mathbf{w}_{\mathbf{y},\tau}, \sigma_k^2 \mathbf{I}\right). \quad (3)$$

The prior distribution of each weight vector $\mathbf{w}_{x,\tau}$ is modeled as

$$\mathbf{p}(\mathbf{w}_{\mathbf{x},\tau}) = \mathcal{N}(\mathbf{w}_{\mathbf{x},\tau} \mid \mathbf{0}, b_\tau \Sigma_p), \quad (4)$$

and where for each task, a conditional one output likelihood GP is modeled with mean $\mathbf{w}_{\mathbf{y},\tau}^\top \mathbf{y}_{k,1:\tau-1}$ and covariance $b_\tau \mathbf{K}$ (see Fig. 8d. d), where $[\mathbf{K}]_{k,l} = k(\hat{\mathbf{x}}_k, \hat{\mathbf{x}}_l)$.

To solve for primal weight vectors $\mathbf{w}_{\mathbf{y},\tau}$, authors in [GMG22] define a prior of these parameters with zero mean and identity covariance matrix and infer a posterior all parameters that is equal to the formulation presented in the standard GPR formulation [WR06]

$$\mathbf{p}([\mathbf{w}_{\mathbf{x},\tau}, \mathbf{w}_{\mathbf{y},\tau}] \mid \Phi, \mathbf{Y}_{1:\tau}) = \mathcal{N}\left(\begin{bmatrix} \mathbf{w}_{\mathbf{x},\tau} \\ \mathbf{w}_{\mathbf{y},\tau} \end{bmatrix} \mid \begin{bmatrix} \bar{\mathbf{w}}_{\mathbf{x},\tau} \\ \bar{\mathbf{w}}_{\mathbf{y},\tau} \end{bmatrix}, \mathbf{A}_\tau^{-1}\right). \quad (5)$$

The posterior is proportional to the product of the prior times the model likelihood. Solving for matrix \mathbf{A} and vector $\bar{\mathbf{w}}_{y,\tau}$ gives the solution, which has to be obtained through a dual formulation provided that the observations are transformed into space \mathcal{H} .

After this, parameters b_τ, σ_k^2 and the kernel parameters are solved by maximizing the joint log-likelihood over all tasks. Once this optimization is done, the solution for $\mathbf{w}_{x,\tau}$ is given in dual form as $\bar{\mathbf{w}}_{x,\tau} = \Phi \alpha_\tau$, where

$$\alpha_\tau = \mathbf{K}_{\mathbf{x},\tau}^{-1} (\mathbf{y}_\tau - \bar{\mathbf{w}}_{\mathbf{y},\tau}^\top \mathbf{Y}_{1:\tau-1}). \quad (6)$$

In the equation, $\mathbf{K}_{x,\tau} = (b_\tau \mathbf{K} + \sigma_\tau^2 \mathbf{I})$, and \mathbf{K} is the kernel matrix containing the kernel dot products $k(\mathbf{x}_k, \mathbf{x}_l)$ between samples.

2. The ensemble method include Bayesian regression and sparse feature selection. It would be helpful to include ablation case study to showcase: (1) performance of Bayesian models with and without sparse feature selection; (2) impact of joint and independent modeling; (3) contribution of kernel selection.

We agree with the Reviewers that an ablation study in this article is of great interest, and we indeed performed it, but the experiments represent the results in a different way to highlight the potential application to power systems. A complete ablation study would require showing independent analyses for each of our 16 models. Alternatively, we removed paragraphs that convey marginal points about ablation studies and added the following paragraphs to the discussion to summarize the conclusions extracted from them; see page 16 of the marked manuscript and Tables SI2-SI5 of the marked Supplemental Information (SI), which summarize the kernel functions.

We analyze our results to isolate the impacts on the performances of sparse learning, the joint distribution, and the kernel function. We first assess the sparse learning impact on the performance by adjusting the hyperparameters to achieve different sparsity levels (ranging from 10 to 1000). We find that the best models consistently include features in the magnitude order of 100s for electric load and solar and 10s for wind (Fig. 9). However, sparse learning is always necessary to reduce the computational complexity of the problem. Additionally, the Relevance Vector Machine (RVM) is an indicator of when the sparse model is unnecessary. The RVM underperforms because it cannot identify the most relevant features (Fig. 2b and e).

We then validate whether modeling the joint distribution across nodes or features improves performance. The performance increased when incorporating the joint distribution (across energy features at the nodal level) into the Gaussian Process for Regression (NLGPR) for electricity demand with Lasso, and solar generation with Elastic Net (EN), compared to independent models (Fig. 2b and c). We can also confirm this in ZP26, which does not have wind generators (Fig. 2e and f). Additionally, we assess the joint distribution (across nodes for a single energy feature) at the system level (SLGPR) when each feature has a kernel, and find that it provides an advantage for wind generation (Fig. 3l).

Finally, we can isolate the effect of the kernel function on the performances. The performance of wind generation declines because electricity load and solar generation favor a linear kernel (\mathcal{K}_L), which negatively impacts the performance when evaluating net demand at the nodal level (Fig. 2f). Wind generation requires an EN-NLGPR with rational quadratic (\mathcal{K}_{RQ}) or Matérn ($\mathcal{K}_{M_{1.5}}$) kernel (Tables SI2-SI5), indicating the existence of non-linear relations between the input features and wind generation. *Multiple kernel learning* exploits non-linear relations by explicitly defining a transformation for each feature source [GA11], but the current MTGPR library does not support multiple kernel functions [GMG22].

As a consequence, Elastic Net with Bayesian Linear Regression (EN-BLR) performs better overall. Still, when we consider ensembles to take advantage of the joint forecasts and evaluate them with proper scoring rules, we observe that joint forecasts are advantageous in power system applications, such as reserve allocations (Fig. 6e).

3. As far as the reviewer known, in 2024, the Institute of Industrial & Systems Engineers (IISE), in partnership with Pacific Gas and Electric Company (PG&E), hosted an energy forecasting competition focused on nodal regions within CAISO. The reviewer has observed that many SOTA deep learning models—such as Temporal Fusion Transformer, Informer, and TimesNet—were widely adopted for day-ahead probabilistic forecasting in that context. It would be helpful for the authors to benchmark their approach against these models as well, particularly

given their growing adoption by system operators and ISOs.

We greatly appreciate that the reviewers shared this insight with us. We were not aware of this article or the challenge.

The IISE PG&E Energy Analytics Challenge 2024 [Azi+25] focused on electricity price forecast at the nodal level. In contrast, we propose a method to dynamically allocate reserves day-ahead, based on a joint probabilistic forecast of demand, solar, and wind generation at the nodal and system levels. The scope of the problem and the source of data are considerably different.

Indeed, some participants used forecasted weather features. However, the documentation lacks a source of numerical weather forecasts and a full description of the weather features used, which is fundamental for comparing performances between methods. Additionally, the methods were not probabilistic, and the performance evaluation was not based on proper scoring rules. Their approach is fundamentally different from the one studied in this investigation.

Deep learning methods are used in energy forecasting problems [Bou+23; Qu+24; Yan+25], but the proposed methodologies generally forecast a single energy feature (demand, solar, or wind), do not provide forecasting uncertainty, and cannot draw predictive scenarios. This method aims to overcome these shortcomings. Therefore, it is challenging to find an existing method that is fully comparable (up to the authors' knowledge).

We expanded the following paragraph in the manuscript to clarify, for readers with the same question, why comparisons are difficult (see page 4 of the marked manuscript).

[...] More recently, deep learning methods based on Temporal Fusion Transformers [Bou+23; Bou+24], Informers [Qu+24], and TimesNets [Yan+25] were used in energy forecasts. However, the proposed methodologies generally forecast a single energy feature (demand, solar, or wind) and do not provide a predictive multivariate density function to draw predictive scenarios that preserve the time structure for risk assessment applications.

4. For reproducibility, the authors provide a GitHub repository; however, the reviewer was unable to locate the hyperparameter settings used in the experiments.

We certainly agree with the reviewers' comment. We addressed this comment together with comment 9. Notice that reviewer no. 1 in comment 6 also had a similar suggestion. We considerably expanded the documentation in the README.md file and add comments in the main scripts referencing the corresponding sections in the manuscript.

In particular, we added comments pointing to the hyperparameters in validation scripts: `val_shallow_learning_batch.py` and `val_multisource_shallow_learning.py`. `val_shallow_learning_batch.py` is used for the cross-validation of the system-level independent/jointly demand, solar, or wind generation, while `val_multisource_shallow_learning.py` is for the cross-validation of the node-level independent/jointly demand, solar, and wind generation.

We also release the GitHub repository with the results visualization (https://github.com/gterren/caiso_power_viz) and shared the results and datasets on a Zenodo data repository [TDM25]. We provided this information in Section Code Availability and Data Availability on page 29 of the marked manuscript.

5. The reviewer believes the claim of “scalability to other ISOs” should be tempered unless the framework is validated on at least a couple of additional systems. CAISO is characterized by a uniquely high penetration of solar PV, pronounced diurnal demand patterns, and relatively mature data transparency and forecasting infrastructure. These features may not generalize to other ISO/RTOs, where renewable integration is often more wind-dominated, load patterns vary seasonally, and the availability and spatial granularity of weather and market data can differ significantly.

We thank the reviewers for sharing their views. We modified the statements in the abstract to temper our claims (see marked manuscript). We did not intend to exaggerate; we hope that, after this correction, the abstract is in agreement with the results.

[...] Improving short-term day-ahead forecasts of renewable energy generation and demand is crucial for system operators to reduce the risk of forecasting errors, coordinate electricity market

operations, and minimize the cost of maintaining power system reliability Here, we introduce and analyze multiple day-ahead forecast models of electricity demand and generation from solar and wind sources based on the joint predictive probability distribution to characterize system-level uncertainty in demand and supply with publicly available weather forecasts. We combine four *sparse learning* methods that identify relevant weather variables with four *Bayesian learning* methods that quantify the uncertainty in the forecast and evaluate each combination of these methods using *proper scoring rules*. Applying these models to the three zones of the California Independent System Operator, we find that the best model combination improves the system operator’s forecast by a skill score of 25.2%. [...]

6. It would be helpful to clarify if the hyperparameters are optimized separately for each day or fixed across the test set?

We appreciate the suggestion. We certainly did not specify that. We added the following text to the manuscript for clarification (see page 28 of the marked manuscript).

[...] The hyperparameter’s cross-validation was performed on the training set, implementing a k -fold cross-validation method. The number of folds was set to $k = 5$ to limit the computational time. After the cross-validation, the model is trained using the entire training set with a fixed set of optimal hyperparameters. The optimal hyperparameter selection criterion (i.e., model selection) is the lowest ES in cross-validation. The performances are evaluated on the testing set [...]

7. It would be helpful to clarify how the group lasso is applied across grouped spatial features.

We thank the reviewers for mentioning this problem. We added on page 21 of the marked manuscript the following paragraph to explain how the features are grouped in the Group Lasso model.

This model is an extension of the Lasso with grouped covariates [YL06]. This model is an extension of the Lasso with grouped covariates [YL06]. We apply the Lasso regularization (L_1 -norm) to all model coefficients, and the group regularization (L_1 -norm) to the coefficients grouped by weather features in a location (i.e., coordinates pairs). The L_2 -norm group regularization is not squared, which makes the penalty non-differentiable at zero, enabling the group variable selection. Its optimization problem is [...]

8. Some minor comments: In the supplementary document, it would be helpful to include a table summarizing the kernel parameters and their respective roles. Additionally, while the Logarithmic Score is introduced in the supplementary material, the reviewer did not observe its use in the main manuscript beyond its mention in the skill score calculation.

We appreciate the reviewers’ suggestion. We added the following table with kernel hyperparameters and their descriptions to the Supplementary Information (SI) (see Table SI1 on the marked supplementary information):

We also removed the Logarithmic Score from (see page 7 of the marked supplementary information).

9. Remarks on code availability: For reproducibility, the authors provide a GitHub repository; however, the reviewer was unable to locate the hyperparameter settings used in the experiments. The readme file in https://github.com/gterren/caiso_power only includes package information.

We certainly agree with the reviewers’ comment and addressed it together with comment 4 (see above). We hope that the response adequately addresses the reviewers’ concerns.

Table 1: Summary of the different parameters in each kernel function and their respective roles.

Kernel	Function	Type	Parameters	Description
Linear (L)	$\mathcal{K}_L(\mathbf{x}_i, \mathbf{x}_j)$	Non-stationary	scale (θ_1) and bias (θ_2)	Low complexity linear transformation with scale (θ_1) and bias parameters (θ_2) —alternative formulation to no-scale and no-bias
Polynomial (P^n)	$\mathcal{K}_{P^n}(\mathbf{x}_i, \mathbf{x}_j)$	Non-stationary	Amplitude (θ_3), bias (θ_4), and polynomial order (n)	An efficient polynomial expansion formulation that avoid its explicit computation in a Hilbert space
Radial Basis Function (RBF)	$\mathcal{K}_{RBF}(\mathbf{x}_i, \mathbf{x}_j)$	Stationary	Length-scale (θ_5) and bias (θ_6)	Non-linear kernel based on similarity metric between samples tuned by the length-scale parameter (as $\theta_5 \rightarrow \infty$ the RBF concentrates the weight on a single sample)
Rational Quadratic (RQ)	$\mathcal{K}_{RQ}(\mathbf{x}_i, \mathbf{x}_j)$	Stationary	Scale (θ_7), length-scale (θ_8), and bias (θ_9)	Scaled mixtures of RBF kernels (as $\theta_7 \rightarrow \infty$ the RQ converges to a RBF kernel)
Matérn (M_ν)	$\mathcal{K}_{M_\nu}(\mathbf{x}_i, \mathbf{x}_j)$	Stationary	Length-scale (θ_{10}), bias (θ_{11}), and Gamma function order (ν)	Improves smoothness control over RBF ($\nu = 0.5$ is an exponential kernel —less smooth—, and as $\nu \rightarrow \infty$ converges to a RBF kernel —more smooth)

Reviewer no. 3

The authors present a novel approach to joint probabilistic day-ahead forecasting of electricity demand, wind and solar. The manuscript is easy to follow and the contribution seems sound, but I’m no expert in these methods. I will therefore stick the following highlevel remarks:

1. The quality of the figures must be improved. Many labels are hard to read, and figures are often overloaded with information. Please relegate non-essential information to the appendix.

We agree with the reviewers’ suggestions. We modified Fig. 1 to make it clearer and increase the font of the labels. In Fig. 2, we removed the kernel labels, relocated box plots to the SI (see Fig. SI8 and 9), marked the best model with an asterisk, and also increased the font size. In addition, we included tables with the numerical results in the SI (see Table SI2-5). We improved the readability of Fig. 4 (see the responses to your questions 2, 3, and 4).

2. In Fig 1, you’re plotting wind and solar variations over seasons, but what has this to do with probabilistic operating reserve sizing? It’s day-ahead forecast errors that drive the need for operating reserves.

We thank the reviewer for the comment. We also consider this variation over seasons unnecessary for this work, and we simplified the plot to show only the annual variation on the timeseries (see Fig. 1). We also increased the font in the plots, and modified the caption as follows:

Electricity demand variability experienced by PG&E (a), SCE (b), and SDG&E (c). Solar generation variability in trading hub NP15 (d), SP15 (e), and ZP26 (f). Wind generation variability in the trading hub NP15 (g) and SP15 (h). The gray regions represent the spread of the 95% Confidence Interval (CI), and the lines are the average hourly demand or generation in winter, spring, summer, and fall of 2020. (i) The map shows which areas’ solar and wind capacity corresponds to a trading hub managed by CAISO (adapted from oasis.caiso.com). (j) Areas served by three major utilities (PG&E, SCE, and SDG&E) in CAISO’s electricity market. (k) Hourly errors in the CAISO’s day-ahead have an increasing trend for wind and solar generation and electricity demand, but also net demand. (l) The energy exchanged by CAISO in the energy imbalance market is increasing, particularly exports [AF23]. (m) The average ancillary services capacity requirements in CAISO are increasing every year [Mar24; Mar21]. Regulation down and non-spinning reserves had a major increase, while spinning reserves were the only product that decreased in 2023.

Additionally, we change the order of the plots in the figure and reference the figure in another place in the introduction to better illustrate the motivation behind the plots.

[...] However, increasing electricity demand and weather-dependent energy sources, as wind and solar, add variability and uncertainty to both demand (Fig. 1a-c) and supply (Fig. 1d-h), increasing the challenges for power system operators to forecast these resources [...]

3. Figures 2 and onwards contain results from many methods. Would it be possible to select the best performing, and relegate the comparison to the supplementary material? This would help the readability.

We removed kernel labels from Figs. 2 and 4, marked the best model with an asterisk, and increased the fonts. We also added the following tables (see Table 2-5) containing the numerical results to the SI (see Tables SI2-SI5). We hope these changes have improved the readability.

Table 2: This table contains the results of Forecast Skill Scores (SS) for independent (Fig. 2b) and joint (Fig. 2e) day-ahead forecasts for each energy feature aggregated across nodes (SS \uparrow is better).

	Kernel	Load	Solar	Wind
Lasso				
BLR		2.36	12.13	-1.48
RVM		-8.8	-4.67	-1.55
GPR	$\mathcal{K}_L, \mathcal{K}_L, \mathcal{K}_{RQ}$	1.17	10.99	3
SLGPR	$\mathcal{K}_L, \mathcal{K}_L, \mathcal{K}_{M1.5}$	-7.68	11.8	4.67
NLGPR	$\mathcal{K}_L, \mathcal{K}_L, \mathcal{K}_L$	6.13	15.79	0.75
OMP				
BLR		-0.01	9.48	-1.71
RVM		-13.38	-0.68	-1.18
GPR	$\mathcal{K}_L, \mathcal{K}_L, \mathcal{K}_{RQ}$	-0.04	12.49	1.58
SLGPR	$\mathcal{K}_L, \mathcal{K}_L, \mathcal{K}_{M2.5}$	2.73	9.8	1.47
NLGPR	$\mathcal{K}_L, \mathcal{K}_L, \mathcal{K}_L$	4.55	13.07	1.17
Elastic Net				
BLR		4.15	12.17	-1.77
RVM		-9.95	1.09	0.58
GPR	$\mathcal{K}_{RQ}, \mathcal{K}_L, \mathcal{K}_{RQ}$	-17.97	11.66	4.95
SLGPR	$\mathcal{K}_L, \mathcal{K}_L, \mathcal{K}_{M1.5}$	-9.56	11.86	5.94
NLGPR	\mathcal{K}_L	5.84	16.77	1.9
Group Lasso				
BLR		-14.02	13.73	-1.44
RVM		-16.5	10.54	-3.75
GPR	$\mathcal{K}_{RQ}, \mathcal{K}_L, \mathcal{K}_{RQ}$	-22.3	12.79	4.53
SLGPR	$\mathcal{K}_L, \mathcal{K}_L, \mathcal{K}_{M1.5}$	-15.36	12.05	5.14
NLGPR	\mathcal{K}_L	-7.1	13.43	-7.21

Table 3: This table contains the results of net demand independent (Fig. 2e) and joint (Fig. 2f) probabilistic forecast SS at nodal and system levels (SS \uparrow is better).

	Kernel	CAISO	NP15	SP15	ZP26
Lasso					
BLR		22.92	19.25	13.88	7.85
RVM		9.96	1.69	-1.99	-16.78
GPR	$\mathcal{K}_L, \mathcal{K}_L, \mathcal{K}_{RQ}$	21.8	19.18	12.32	6.98
SLGPR	$\mathcal{K}_L, \mathcal{K}_L, \mathcal{K}_{M1.5}$	21.04	13.38	14.76	9.58
NLGPR	\mathcal{K}_L	24.81	16.6	13.79	8.65
OMP					
BLR		23.5	11.58	16.67	9.15
RVM		13.92	-0.1	0.79	-7.05
GPR	$\mathcal{K}_L, \mathcal{K}_L, \mathcal{K}_{RQ}$	24.25	11.69	16.57	8.96
SLGPR	$\mathcal{K}_L, \mathcal{K}_L, \mathcal{K}_{M2.5}$	23.43	14.23	16.3	9.94
NLGPR	\mathcal{K}_L	23.11	12.44	13.61	10.14
Elastic Net					
BLR		25.23	19.4	16.38	9.02
RVM		11.23	0.66	0.7	-12.01
GPR	$\mathcal{K}_{RQ}, \mathcal{K}_L, \mathcal{K}_{RQ}$	9.73	4.03	6.26	-0.25
SLGPR	$\mathcal{K}_L, \mathcal{K}_L, \mathcal{K}_{M1.5}$	20.7	12.43	15.34	9.22
NLGPR	\mathcal{K}_L	24.39	16.3	12.86	9.26
Group Lasso					
BLR		18.62	5.9	12.65	9.44
RVM		12.13	-13.66	3.15	-8.21
GPR	$\mathcal{K}_{RQ}, \mathcal{K}_L, \mathcal{K}_{RQ}$	9.16	1.21	6.05	-5.39
SLGPR	$\mathcal{K}_L, \mathcal{K}_L, \mathcal{K}_{M1.5}$	19	7.75	13.46	9.04
NLGPR	\mathcal{K}_L	12.99	0.83	2.82	7.19

4. The application to reserve sizing does not add value. The whole comparison hinges on some - in my opinion, questionable - assumptions on how to treat import/export. I would remove this section, and focus on the probabilistic

Table 4: Scores achieved by the different day-ahead probability forecasting models of electricity demand (Fig. 5a, d, and g), solar generation (Fig. 5b, e, and h), and wind generation (Fig. 5c, f, and i). The scores are Energy Score (ES), Variogram Score ($VS^{0.5}$), Interval Score (IS), Skill Score (SS_{RMSE}) compared to CAISO’s forecast, and computational time.

Sparse Learning	Dense Learning	Kernel	Feature	RMSE	FS	ES	VS	IS	Tr. Time	Ts. Time
Lasso	BLR		load	1104.12	2.36	39.75	2380.29	8163.84	116.70	0.80
Lasso	BLR		solar	616.20	12.13	33.63	2214.89	5454.18	1070.64	2.94
Lasso	BLR		wind	449.21	-1.48	31.78	2330.46	4809.18	113.95	0.85
Lasso	GPR	\mathcal{K}_L	load	1117.54	1.17	40.40	2465.06	8324.47	834.35	0.99
Lasso	GPR	\mathcal{K}_L	solar	624.19	10.99	33.00	2279.55	5452.95	3538.41	2.48
Lasso	GPR	\mathcal{K}_{RQ}	wind	429.37	3.00	31.53	2186.33	4980.05	3289.84	0.92
Lasso	RVM		load	1230.27	-8.80	48.83	3633.57	20340.11	1122.05	1.30
Lasso	RVM		solar	734.03	-4.67	37.52	2943.02	7762.18	644.41	0.84
Lasso	RVM		wind	449.50	-1.55	32.24	2731.55	4967.20	29.49	0.81
Lasso	SLGPR	$\mathcal{K}_{M_{1.5}}$	wind	422.00	4.67	31.37	2182.85	5668.83	2942.37	5.45
Lasso	SLGPR	\mathcal{K}_L	load	1217.63	-7.68	42.64	2505.08	11011.55	3346.59	11.29
Lasso	SLGPR	\mathcal{K}_L	solar	618.52	11.80	33.15	2117.53	6760.84	6511.18	10.05
OMP	BLR		load	1130.85	-0.01	41.93	2654.42	8428.81	19.47	0.89
OMP	BLR		solar	634.76	9.48	33.65	2329.27	5415.37	45.59	1.17
OMP	BLR		wind	450.21	-1.71	31.82	2334.30	4805.96	15.02	1.00
OMP	GPR	\mathcal{K}_L	load	1131.19	-0.04	42.31	2708.22	8385.01	503.95	0.93
OMP	GPR	\mathcal{K}_L	solar	613.68	12.49	32.65	2351.23	4628.44	896.46	0.95
OMP	GPR	\mathcal{K}_{RQ}	wind	435.65	1.58	32.15	2229.28	4930.26	4033.00	1.13
OMP	RVM		load	1282.02	-13.38	62.00	5704.26	14714.92	112.41	0.81
OMP	RVM		solar	706.04	-0.68	37.23	2910.84	6972.10	23.14	0.82
OMP	RVM		wind	447.87	-1.18	32.43	2791.95	4883.68	2.59	0.80
OMP	SLGPR	$\mathcal{K}_{M_{2.5}}$	wind	436.16	1.47	32.56	2253.62	5720.02	2468.53	4.82
OMP	SLGPR	\mathcal{K}_L	load	1099.94	2.73	40.45	2483.98	9424.48	3251.94	11.08
OMP	SLGPR	\mathcal{K}_L	solar	632.53	9.80	33.50	2119.39	7060.30	6817.35	17.01
Elastic Net	BLR		load	1083.79	4.15	39.32	2394.04	8087.04	80.52	0.81
Elastic Net	BLR		solar	615.89	12.17	33.97	2197.84	5444.52	913.85	5.84
Elastic Net	BLR		wind	450.48	-1.77	31.91	2336.52	4820.43	148.19	1.08
Elastic Net	GPR	\mathcal{K}_{RQ}	load	1333.92	-17.97	43.05	2534.89	18705.54	6420.78	1.26
Elastic Net	GPR	\mathcal{K}_L	solar	619.51	11.66	33.03	2262.97	5197.78	4590.76	3.12
Elastic Net	GPR	\mathcal{K}_{RQ}	wind	420.76	4.95	30.83	2135.10	4942.62	2860.59	0.90
Elastic Net	RVM		load	1243.30	-9.95	49.50	3697.87	20951.83	1252.71	1.47
Elastic Net	RVM		solar	693.65	1.09	35.87	2747.27	7061.39	1401.97	1.18
Elastic Net	RVM		wind	440.09	0.58	31.87	2679.50	4899.63	28.87	0.72
Elastic Net	SLGPR	\mathcal{K}_L	load	1238.88	-9.56	42.99	2540.53	11531.38	3520.53	11.55
Elastic Net	SLGPR	\mathcal{K}_L	solar	618.11	11.86	33.98	2177.11	6447.40	11310.36	11.19
Elastic Net	SLGPR	$\mathcal{K}_{M_{1.5}}$	wind	416.34	5.94	30.75	2144.09	5615.62	2622.51	4.58
Group Lasso	BLR		load	1289.34	-14.02	47.43	3088.95	9637.66	3943.20	0.76
Group Lasso	BLR		solar	604.97	13.73	32.49	2211.55	4829.10	7353.41	1.43
Group Lasso	BLR		wind	449.04	-1.44	31.87	2358.61	4847.03	558.66	0.93
Group Lasso	GPR	\mathcal{K}_{RQ}	load	1382.91	-22.30	44.16	2680.14	19057.39	9022.40	0.99
Group Lasso	GPR	\mathcal{K}_L	solar	611.54	12.79	32.18	2275.40	5153.34	6159.04	1.25
Group Lasso	GPR	\mathcal{K}_{RQ}	wind	422.61	4.53	30.99	2135.04	4716.95	3831.63	0.80
Group Lasso	RVM		load	1317.32	-16.50	56.15	4580.24	20546.85	5210.57	0.94
Group Lasso	RVM		solar	627.34	10.54	34.97	2776.67	4888.73	5058.80	0.74
Group Lasso	RVM		wind	459.27	-3.75	32.94	2836.01	4979.35	581.40	0.72
Group Lasso	SLGPR	\mathcal{K}_L	load	1304.44	-15.36	47.26	2950.27	11824.63	6883.39	9.83
Group Lasso	SLGPR	\mathcal{K}_L	solar	616.73	12.05	33.53	2153.48	6618.87	11917.14	9.47
Group Lasso	SLGPR	$\mathcal{K}_{M_{1.5}}$	wind	419.90	5.14	31.06	2158.79	5664.73	4482.32	8.28

forecasting. The contribution and relevance thereof is sufficient.

We thank the reviewer for sharing their concern. Reviewer no. 1 in question 4 also shared a similar concern. We modified the labels in Fig. 6 (see page 14 of the marked manuscript). We defined over-forecast events as exports or variable renewable energy curtailment and under-forecast events as imports or demand shedding. Because of the zonal model, we do not estimate congestion. We hope this modification is sufficient, since we do not quantify the cost of these events. In addition, we add the following changes to the caption:

[...] Net demand forecast reserves’ upper bound (*ub*) and lower bound (*lb*) shown by lower solid and upper dashed lines and reserves (*rs*) shown by shaded areas between solid and dashed lines—the reserves’ upper and lower bounds are the confidence interval lower and upper bounds for OMP-NLGPR (μ). (c) Upward + downward reserves allocation for CAISO and OMP-NLGPR (μ) forecasts. (d) Energy imports (*im*) (and demand shedding if energy is not available in the regional energy imbalance market) and exports (and wind or solar curtailment, *ct*, if demand is lacking in the energy imbalance market) resulting from the CAISO and the proposed allocation methodology using OMP-NLGPR (μ) forecast (positive means energy imports required, while negative means energy exports available). (e) Average imports and energy curtailment obtained from CAISO’s forecast and the proposed ensemble of probabilistic day-ahead net energy forecasts across the sim-

Table 5: Scores achieved by the proposed forecasting models when evaluated at the system level (Fig. 5a-e): Energy Score (ES), Variogram Score ($VS^{0.5}$), Interval Score (IS), Skill Score (SS_{RMSE}) compared to CAISO’s forecast, and computational time.

Sparse Learning	Dense Learning	Kernel	RMSE	FS	ES	VS	IS	Tr. Time	Ts. Time
Lasso	BLR		1349.56	22.92	86.61	21845.28	14740.34	1301.29	4.59
Lasso	RVM		1576.58	9.96	100.97	28707.23	26310.88	1795.95	2.95
Lasso	GPR	$\mathcal{K}_L, \mathcal{K}_L, \mathcal{K}_{RQ}$	1369.25	21.8	86.78	21871.29	15184.85	7662.6	4.39
Lasso	SLGPR	$\mathcal{K}_L, \mathcal{K}_L, \mathcal{K}_{M1.5}$	1382.44	21.04	89.61	21528.81	17575.56	12800.15	26.79
Lasso	NLGPR	\mathcal{K}_L	1316.5	24.81	85.34	21800.45	14016.98	10817.48	24.47
OMP	BLR		1339.45	23.5	88.98	22692.35	14413.46	80.08	3.06
OMP	RVM		1507.14	13.92	115.77	32591.66	20659.4	138.14	2.42
OMP	GPR	$\mathcal{K}_L, \mathcal{K}_L, \mathcal{K}_{RQ}$	1326.29	24.25	88.32	22467.54	14304.54	5433.4	3
OMP	SLGPR	$\mathcal{K}_L, \mathcal{K}_L, \mathcal{K}_{M1.5}$	1340.6	23.43	87.62	21694.01	16650.82	12537.82	32.9
OMP	NLGPR	\mathcal{K}_L	1346.3	23.11	87.02	22429.9	14488.85	9557.45	24.51
Elastic Net	BLR		1309.16	25.23	85.94	21667.17	14009.14	1142.57	7.72
Elastic Net	RVM		1554.29	11.23	100	27840.94	25955.55	2683.55	3.38
Elastic Net	GPR	$\mathcal{K}_{RQ}, \mathcal{K}_L, \mathcal{K}_{RQ}$	1580.55	9.73	91.71	22264.07	26499.03	13872.13	5.27
Elastic Net	SLGPR	$\mathcal{K}_L, \mathcal{K}_L, \mathcal{K}_{M1.5}$	1388.48	20.7	90.37	21558.69	17739.86	17453.4	27.33
Elastic Net	NLGPR	$\mathcal{K}_L, \mathcal{K}_L, \mathcal{K}_L$	1323.91	24.39	85.46	21815.17	14128.5	10989.06	24.15
Group Lasso	BLR		1424.91	18.62	94.18	23506.19	15456.15	11855.26	3.12
Group Lasso	RVM		1538.54	12.13	107.33	29836.69	22977.3	10850.77	2.39
Group Lasso	GPR	$\mathcal{K}_{RQ}, \mathcal{K}_L, \mathcal{K}_{RQ}$	1590.48	9.16	92.23	22659.52	25706.81	19013.06	3.05
Group Lasso	SLGPR	$\mathcal{K}_L, \mathcal{K}_L, \mathcal{K}_{M1.5}$	1418.19	19	95.12	22593.11	17671.88	23282.86	27.57
Group Lasso	NLGPR	\mathcal{K}_L	1523.46	12.99	92.54	24772	16861.12	79001.78	29.02

ulated period from May 2022 to Feb. 2023 (test samples). White dots represent the net balance between imports, exports, and energy curtailment.

We considerably modified the wording in the entire Section Probabilistic Day-Ahead Energy Forecast for Reserves Allocation (see page 13-16 of the marked manuscript), and modified the first paragraph as follows:

[...] When positive net demand forecast errors (generation overestimation or demand underestimation) exceed upward reserves, [GGF22], ISOs import energy from neighboring interconnected regions or, in extreme cases, shed demand. When negative net demand forecast errors (generation underestimation or demand overestimation) exceed downward reserves, ISOs export electricity to neighboring regions or curtail VRE generation. In California, the CAISO imports from and exports to the Western Energy Imbalance Market [AF23]. [...]

We rather prefer to maintain this figure as a simple illustration of a potential application in a power system. We added some limitations to the discussion section to caution the reader about potential limitations (see marked manuscript’s page 18).

The proposed dynamic reserves allocation serves as an example of how to utilize a probabilistic forecast in power system operations. [...]

We intentionally integrated the ability to generate predictive scenarios associated with probabilities in a day-ahead forecast to assess the Conditional Value at Risk (CVaR) on dynamic reserves allocation and analyze the impact of spatial correlations on both demand and generation on grid congestion, considering congestion and committed units to account for energy imports and exports more accurately. However, this investigation will require methods and datasets beyond the scope of this study.

References

- [WR06] Christopher KI Williams and Carl Edward Rasmussen. *Gaussian processes for machine learning*. Vol. 2. 3. MIT press Cambridge, MA, 2006.
- [YL06] Ming Yuan and Yi Lin. “Model selection and estimation in regression with grouped variables”. In: *Journal of the Royal Statistical Society Series B: Statistical Methodology* 68.1 (2006), pp. 49–67.
- [GA11] Mehmet Gönen and Ethem Alpaydm. “Multiple kernel learning algorithms”. In: *The Journal of Machine Learning Research* 12 (2011), pp. 2211–2268.
- [Mar21] Department of Market Monitoring. *2020 Annual Report on Market Issues and Performance*. Tech. rep. California Independent System Operator (CAISO), Aug. 2021.
- [GGF22] Ningchao Gao, David Wenzhong Gao, and Xin Fang. “Manage Real-Time Power Imbalance with Renewable Energy: Fast Generation Dispatch or Adaptive Frequency Regulation?” In: *IEEE Transactions on Power Systems* (2022).
- [GMG22] Óscar García-Hinde, Manel Martínez-Ramón, and Vanessa Gómez-Verdejo. “A conditional one-output likelihood formulation for multitask Gaussian processes”. In: *Neurocomputing* 509 (2022), pp. 257–270.
- [AF23] Market Analysis and Forecasting. *Western Energy Imbalance Market Benefits*. Tech. rep. California Independent System Operator (CAISO), 2023.
- [Bou+23] Oussama Boussif et al. “Improving day-ahead solar irradiance time series forecasting by leveraging spatio-temporal context”. In: *Advances in Neural Information Processing Systems* 36 (2023), pp. 2342–2367.
- [Bou+24] Oussama Boussif et al. “Improving day-ahead Solar Irradiance Time Series Forecasting by Leveraging Spatio-Temporal Context”. In: *Advances in Neural Information Processing Systems* 36 (2024).
- [Mar24] Department of Market Monitoring. *2023 Annual Report on Market Issues and Performance*. Tech. rep. California Independent System Operator (CAISO), July 2024.
- [Qu+24] Kai Qu et al. “Forwardformer: Efficient Transformer With Multi-Scale Forward Self-Attention for Day-Ahead Load Forecasting”. In: *IEEE Transactions on Power Systems* 39.1 (2024), pp. 1421–1433. DOI: 10.1109/TPWRS.2023.3266369.
- [Azi+25] Ahmed Aziz Ezzat et al. “IISE PG&E Energy Analytics Challenge 2024: Forecasting day-ahead electricity prices”. In: *IISE Transactions* (2025), pp. 1–13.
- [TDM25] Guillermo Terren-Serrano, Ranjit Deshmukh, and Manel Martinez-Ramon. *Datasets & Results for Probabilistic Day-Ahead Forecasting of System-Level Renewable Energy and Electricity Demand*. Version 1. Aug. 2025. DOI: 10.5281/zenodo.16729434. URL: <https://doi.org/10.5281/zenodo.16729434>.
- [Yan+25] Mao Yang et al. “A framework of day-ahead wind supply power forecasting by risk scenario perception”. In: *IEEE Transactions on Sustainable Energy* (2025).